# Reductions in wind farm main bearing rating lives resulting from wake impingement

Julian Quick[1], Edward Hart[2], Marcus Binder Nilsen[1], Rasmus Sode Lund[1], Jaime Liew[1], Piinshin Huang[1], Pierre-Elouan Rethore[1], Jonathan Keller[3], Wooyong Song[4], and Yi Guo[5]

[1]Technical University of Denmark, Lyngby, Denmark
[2]University of Strathclyde, Glasgow, UK
[3]National Laboratory of the Rockies, Golden, CO, U.S.A.
[4]Offshore Renewable Energy (ORE) Catapult, Blyth, UK
[5]Department of Mechanical Engineering, Clemson University, North Charleston, SC, U.S.A.

**Correspondence:** Julian Quick (juqu@dtu.dk)

**Abstract.** This paper studies the impacts of wake impingement on main bearing rating lives predicted during the wind turbine design stage. A computational tool chain was developed to explore and quantify these effects across a wind farm populated by 10 MW wind turbines. Wind field and turbine load modelling was undertaken using the Dynamiks Python package, including application of a dynamic wake meandering model. The ISO 281 basic bearing rating life formulation was subsequently applied
in order to evaluate impacts from wake effects. Analyses included a two-turbine parametric analysis, followed by a full wind farm analysis undertaken for the TotalControl 32-turbine reference wind farm, including full wind rose simulations across all operational wind speeds. Site conditions were accounted for using a Weibull wind speed distribution and a range of parametric wind direction rose models. Results indicate that wind farm main bearing rating lives are negatively impacted by the effects of wake impingement, resulting in rating life reductions for the analysed wind farm on the order of 16 % on average and
as much as 20-25 %, both for the locating main bearing. Despite these high sensitivities, it is important to note that these resultant rating lives (i.e. the *predicted* lives) still far exceed the standard wind turbine operational lifetimes of 20–30 years. Wake impacts were also found to be asymmetrically related to the side on which the rotor is impinged, suggesting that, for the main bearing, there may be a "better" side for wake impingement to occur. Rating life sensitivities to wind rose shape were also observed. While these findings must be interpreted with due consideration for the various methodological limitations present,
they provide compelling evidence that wake effects at the wind farm level should necessarily be included when undertaking main bearing operational load modelling, rating life assessment, or other load-related analyses.

## 1 Introduction

Main bearing failures remain a significant reliability challenge within the wind industry, with recent high-volume studies indicating only around half of a main bearing population's minimum design life[1] (20 years) may be attained in practice (Hart et al.,
2023; EPRI, 2024). As will be elaborated on below, there is therefore a significant gap between the main bearing (predicted)

---

[1]*Life* in the context of bearings refers to the time at which 10 % of the population has failed.

rating life and (observed) field life. It is anticipated that main bearing failure rates may grow with the increase in turbine rated power (EPRI, 2024), which is consistent with the observation that main bearing loading scales unfavorably with turbine size (Hart et al., 2022). No principal root cause of main bearing field failures has yet been identified; observed damage modes have included those stemming from surface and subsurface initiation, stray currents, lubrication failures, overloading, and improper bearing assembly/fit (Hart et al., 2023; EPRI, 2024). Efforts are therefore ongoing to determine a set of fundamental drivers leading to premature main bearing failures and, in turn, identify strategies to resolve or ameliorate these issues.

Bearing design and selection normally includes the application of ISO standards (ISO 281 (ISO 281:2007, E) and ISO 16281), which seek to estimate lifetime across a population of identical bearings at risk from surface- and subsurface-initiated rolling contact fatigue (RCF). An important caveat to the ISO standards is that they make no claims concerning expected lifetimes for out-of-scope damage modes. Those alternative damage mechanisms tend to be difficult to model and hard to predict, hence the lack of comparable and generalized life assessment formulations. Investigations of potential causes of premature main bearing failures therefore commonly utilize the ISO standards (Zheng et al., 2020; Kenworthy et al., 2024; Krathe et al., 2024; Ishihara et al., 2025). A limitation of such approaches is the resulting focus on surface and subsurface RCF and therefore the exclusion of other potentially important damage modes. However, field observations include damage that may have resulted from RCF (Hart et al., 2023), and it has not been conclusively demonstrated that RCF is not an important factor in main bearing field failures. A key question in this context is therefore: *Can ISO-based main bearing rating life assessment, in conjunction with realistic system[2] modeling, account for reported levels of main bearing failures?* This gives rise to the related question: *What constitutes a sufficiently realistic system model in this context?* Kenworthy et al. (2024) investigated the first question by applying ISO 281 to the main bearing of an individually modelled 1.5 MW wind turbine. While the question was answered in the negative for that setup, the results included high levels of sensitivity to vertical wind shear flow asymmetry. As a result, lateral flow asymmetry caused by wake impingement from an upwind turbine was posited as a possibly overlooked life reduction factor, i.e. a potential insufficiency in the model. The current work seeks to contribute towards addressing both of the above questions, by considering: *For a candidate model wind farm, what is the relative change in main bearing rating life when including the effects of wake impinged operation?*

The remainder of the paper is organized as follows: Sect. 2 presents and discusses relevant background information. Sect. 3 outlines the methods and cases examined in this study. Results are then presented and discussed in Sects. 4 and 5, respectively. Finally, conclusions are presented in Sect. 6.

## 2  Background

The current section outlines relevant theory, prior work, and extant modelling capabilities which are pertinent to and/or utilized within the current study. Sect. 2.1 introduces the ISO 281 bearing rating life assessment standard. Sect. 2.2 reviews previous

---

[2]"System" in this context refers to the various interacting components, processes and phenomena which together determine wind turbine and/or wind farm operational and loading conditions: a bearing, within a drivetrain, within a wind turbine, interacting with atmospheric flow, within a wind farm, etc.

applications of the ISO 281 standard to wind turbine bearings. Sect. 2.3 discusses relevant approaches for modelling wind farms.

## 2.1 ISO 281 bearing rating life assessment

ISO 281 formulations provide an estimate of bearing life across a population of identical rolling bearings. The basic rating life is the life that 90 % of the bearing population is expected to attain or exceed; equivalently, this is the point at which 10 % of the population is expected to have failed. A detailed review and application case study in the context of main bearings was previously provided by Kenworthy et al. (2024). The basic rating life for a radial roller bearing takes the form,

$$L_{10} = \left( \frac{C_{\mathrm{D}}}{P_{\mathrm{eq}}} \right)^{10/3}, \tag{1}$$

where $C_{\mathrm{D}}$ is the basic dynamic load rating, generally supplied by the bearing manufacturer. The dynamic equivalent radial load, $P_{\mathrm{eq}}$, is calculated from the applied radial and axial bearing loads, $F_{\mathrm{r}}$ and $F_{\mathrm{a}}$, as

$$P_{\mathrm{eq}} = X F_{\mathrm{r}} + Y F_{\mathrm{a}}. \tag{2}$$

Coefficient values $X$ and $Y$ are prescribed by ISO 281, and they depend on the bearing nominal contact angle and the load ratio $F_{\mathrm{a}}/F_{\mathrm{r}}$. ISO equations give $L_{10}$ values in millions of revolutions; these are then readily converted to units of time using the rotational speed of the wind turbine low-speed shaft.

The above equation pertains to the case of a constant applied-load combination, $F_{\mathrm{r}}$ and $F_{\mathrm{a}}$. Variable operating conditions, such as those experienced by wind turbine bearings, are accommodated using an assumption of linear damage accumulation for consumed bearing life. Under this assumption, the resultant rating life, $L_{\mathrm{res}}$, for a bearing operated in $n$ different sets of conditions with associated rating lives $L_1, \ldots, L_n$ takes the form of a harmonic mean (Kenworthy et al., 2024),

$$L_{\mathrm{res}} = \frac{1}{\frac{\phi_1}{L_1} + \frac{\phi_2}{L_2} + \ldots + \frac{\phi_n}{L_n}}, \tag{3}$$

where $\phi_i$ is the proportion of total operation that occurred under operating-condition $i$. The combination of condition-specific rating lives into a resultant life may be undertaken in stages. The multistage approach has been shown to be equivalent to a single simultaneous combination of all cases (Kenworthy et al., 2024). This provides a convenient and staged approach to resolving the resultant bearing rating life in practice.

While both the *modified* rating life of ISO 281 and the enhanced formulations of ISO 16281 seek to account for more detailed factors – such as contamination, the bearing internal load distribution, clearance and misalignment – additional technical and environmental data are required. It has also been shown that the load distribution within large main bearings is highly sensitive to model fidelity, with the influence of bearing housing and turbine bedplate deflections playing an important role (Kock et al., 2019). As a result, and given the focus of the current study is quantifying the *relative* impacts of wake impingement, the ISO 281 basic rating life formulation was applied.

## 2.2 Main bearing fatigue life analysis

A number of studies have applied ISO 281 to assess main bearing rating lives. Kenworthy et al. (2024), discussed previously, studied the rating life prediction for an individually modeled 1.5 MW wind turbine main bearing resulting from IEC 61400-1 and ISO 281 design processes. Rating life assessment was carried out for various combinations of bearing temperature, wind field characteristics, lubricant viscosity, and contamination levels; main bearing loading was estimated from aeroelastically derived turbine-hub loading via a static force balance at each time step. The results of this analysis implied the prescribed life assessment process did not account for reported rates of main bearing failures in 1 to 3 MW wind turbines. It was also suggested that the impacts of wake impingement on main bearing rating life should be considered in future work. Ishihara et al. (2025) sought to quantify the role of system inertial loading on main bearing rating life by proposing a novel main bearing "pounding" model which estimates the load augmentation resulting from impact events at bearing interfaces. The potential influence on RCF life was assessed by calculating the mean inertial increase in bearing loading and the impacts of bearing clearance and passing these adjusted values into ISO 281 rating life equations. Their inclusion of inertial loading and bearing clearance significantly reduced the bearing $L_{10}$ rating life from 144 years down to 39 years (accounting for clearance only) or 9.7 years (accounting for inertial loading and clearance). While their findings and the presented model are interesting, there are both conceptual and implementation issues present which make it difficult to determine the validity of their results. First, "pounding" of the main bearing is described as beginning once internal wear results in increased bearing clearance. However, if wear is present to this extent, then the bearing has already failed or is at least in the process of failing. Furthermore, wear is precisely one of the out-of-scope damage modes for which ISO 281 claims not to apply. Second, the pounding model clearance (which determines the gap across which the shaft can accelerate) was set by manually measuring the bearing's radial clearance uptower while the turbine was nonoperational. This is problematic, since the main bearing in question is a three-point-mounted double-row spherical roller bearing which reacts to both axial (thrust) and radial loads. Turbine thrust loading results in downwind displacement of the low-speed shaft, with a subsequent force response from the downwind bearing row Hart (2020); Guo et al. (2022); de Mello et al. (2023). Axial displacement of this kind reduces the effective bearing radial clearance during operation. This includes periods of full circumferential loading of the downwind row (Guo et al., 2022), where the effective clearance becomes zero. The value of radial clearance used for the pounding model and "life ratio" of Ishihara et al. (2025) is therefore likely excessive. For the above reasons, it is not clear to what extent the rating life results of this previous study should be considered valid or representative at this stage.

Krathe et al. (2024) investigated the sensitivity of main bearing rating life, per ISO 281, to the choice of turbulence model. The Kaimal and Mann standard kinematic turbulence models were compared, in this context, to wind fields generated via constrained extrapolation of large-eddy simulation (LES) data. A coupled medium-fidelity drivetrain model was developed and implemented in OpenFAST, including verification against a more complex multibody drivetrain model. RCF life consumption based on ISO 281 was found to be fastest for the upwind bearing by 2 orders of magnitude, while the upwind bearing $C_{D}$ value was only 23% greater than that of the downwind bearing, in their four-point drivetrain. Differences between turbulence models were found to be small (2–10 %) for the upwind bearing but more significant (10–40 %) for the downwind bearing. Of the two

standard turbulence models, the Mann model most closely re-created the rating life values obtained using LES extrapolated
inflow data. This work highlights turbulence as an important and open aspect of model sufficiency in main bearing research.
Krathe et al. (2025) analysed main bearing response to wake impingement for a pair of 15-MW floating direct-drive wind
turbines under three atmospheric turbulence conditions, modelled using large eddy simulation for 8 m/s inflow and five rear-
turbine offsets. Asymmetric rating-life consumption impacts were observed with respect to the offset position of the downwind
turbine; with the most "damaging" conditions occurring when the wake partially impinges an upward traveling blade. Rating
life consumption on the locating main bearing was found to be slower for the waked turbine, compared to the unwaked turbine,
in all cases. This was attributed to a reduction in thrust for the waked turbine. While undoubtedly interesting, the study only
considers two turbines and a single wind speed. This was due to the computational expense of the large eddy simulations. In
addition, since floating wind turbines were modeled, it is not possible to delineate which results hold in general and which are
specific to floating turbines.

Other works have implemented ISO 16281 formulations for the purposes of wind turbine main bearing rating life analysis
(Zheng et al., 2020; Jiang et al., 2022a, b); however, these are principally focused on developing the modelling and analysis
capabilities themselves rather than on their application to fundamental questions of failure drivers and expected field lives.

## 2.3   Modelling wind turbines, wind farms and wakes

The wind turbine wake is a downstream region of air with increased vorticity and decreased velocity, sometimes persisting for
several kilometers (Dong et al., 2022). Within a wind farm, practical design and cost considerations generally result in turbines
being sited at a proximity which necessitates them operating within the wake of other turbines a majority of the time. The rapid
evolution of wind turbines in the past few decades towards larger and more powerful machines has resulted in increasingly
complex wake dynamics. The full impacts of wake impingement on downstream turbines are not yet fully understood, but it
is known that impingement can increase structural loads, reduce expected service lives, and limit energy capture (Veers et al.,
2023). Loading from partial wake impingement can influence the optimal wind farm control strategy (Stanley et al., 2020a).
The IEC standard has some provisions for modelling wake impacts on turbines, though these often manifest as enhanced inflow
turbulence without partial wake overlap. To date, analyses of load impacts from wake impingement have generally focused on
turbine blades and towers (Riva et al., 2020; Stanley et al., 2020b; Shaler et al., 2022). Where the main bearing or drivetrain
has been considered, rating life impacts have not been quantified (van Binsbergen et al., 2020), or only the added turbulence
aspect of wakes was modelled (Moghadam et al., 2023), or the analysis was limited by the numbers of turbines and operating
points considered (Krathe et al., 2025).

Wake and large-scale turbine dynamics can be modelled using the Dynamiks Python package (Technical University of
Denmark, 2025). Dynamiks is a modular framework used to simulate dynamic wind farm flows. The dynamic wake meandering
model (Larsen et al., 2008; Liew et al., 2023) is used to model the turbine wakes by tracking wake particles as they convect
downstream of wind turbines. The software is highly modular, allowing for the specification of custom inflow, particle motion,
deficit, deflection, and superposition models. A Dynamiks site is defined in terms of the mean wind field characteristics,
turbulence intensity, shear, and a turbulence field. It is common to model the turbulence field using a turbulence box generated

via the Mann model (Mann, 1998). It can be configured to run simple turbine representations, or it can be coupled with the HAWC2 aeroelastic analysis software for detailed turbine simulations (Larsen and Hansen, 2007; Madsen et al., 2020).

The distribution of wind speed and wind direction at a wind farm will strongly affect both energy yield and component reliability, especially when wake effects are considered. The two-parameter (scale $C$ and shape $k$) Weibull distribution is a standard model for representing the 10-minute mean wind speed distribution (Kenworthy et al., 2024). A typical shape parameter is $k = 2$, and $C$ may be estimated using $k$ and a specified site annual mean wind speed (Gryning et al., 2016). A parametric model for describing the distribution of wind direction at a site has recently been proposed (Hart, 2025). The

parametric model utilises ellipses-of-unit-area to specify wind roses, with the probability associated with any direction-segment being equal to that segment's area. The resulting *generalized elliptical wind direction rose* has three parameters: a prevailing wind direction ($\theta_{\mathrm{prev}}$), elliptical parameter ($a$) and folding parameter ($f$). Restricting the ellipse to be of unit area results in a single parameter, $a$, which determines the shape of the baseline ellipse, from circular to increasingly elongated. The folding parameter then specifies a proportion (from 0 to 1) of probability mass to be "folded" across the ellipse minor-axis, thereby

establishing a chosen level of bi- versus uni-directionality. Finally, the resulting wind rose model is rotated to obtain the specified prevailing wind direction, $\theta_{\mathrm{prev}}$. Together, the above models (Weibull plus elliptical wind rose) for parametrically describing distributions of wind speed and wind direction at a site readily enable resource characterization.

## 3 Methodology

This study was conducted in two principal stages. First, a two-turbine parametric analysis was undertaken to characterize the

165 impacts on main bearing rating lives from a single wake for different levels of impingement and turbine separation. Second, a complete (IEC 61400-1 compliant) wake-inclusive full-wind-rose main bearing rating life assessment was undertaken for all main bearings within a 32-turbine reference wind farm. The current section details the full modelling and analysis tool chain (Sect. 3.1), followed by a description of the two-turbine parametric analysis (Sect. 3.2) and the complete wind farm analysis (Sect. 3.3).

### 3.1 Modelling tool chain

Various modelling tools, introduced in Sect. 2, were utilized and combined in this study. The specifics of their applications within our modelling framework are described below.

**Flow field modelling**: Ambient turbulence was generated via the Mann model using a length scale of 33.6 m, the eddy lifetime parameter set to $\Gamma = 3.9$, and a turbulence intensity of 5 % (a low-to-moderate offshore turbulence level (Marek et al., 2016)). A

175 power-law vertical wind shear profile was applied throughout, with shear exponent 0.2. Wake and large-scale turbine dynamics were modelled using the Dynamiks software package (Technical University of Denmark, 2025), including a dynamic wake meandering model (Larsen et al., 2008; Liew et al., 2023). Wake dynamics approximations used 21 radial segments, a maximum of 30 particles per turbine wake, a minimum travel distance of $0.1D$ (where $D$ is the turbine rotor diameter) and a maximum

radius of 294 m to model the wake profile. Overlapping wake deficits were summed linearly. These flow simulations were run using a time step of 1 s, using the cutoff frequency suggested by Lio et al. (2021). Note, this is a far-wake model which can be considered to be effective from approximately $3D+$ downstream of the wake-producing turbine (Quick et al., 2024).

**Wind turbine simulation**: Within Dynamiks, each wind turbine was modelled using HAWC2 representations of the Technical University of Denmark's (DTU's) 10 MW reference wind turbine (Bak et al., 2013a). This turbine has a hub height of 119 m, a rotor diameter of 178.3 m, and a shaft tilt of 5 degrees. The associated power and thrust-coefficient curves are shown in Figure 1. Hub-centre loading time series in 6 degrees of freedom were extracted from each simulated turbine and applied to main bearing load estimation. The turbines were controlled using the DTU Basic Controller (Hansen and Henriksen, 2013). These turbine simulations were run with a time step of 0.01 s. The individual turbine wind fields were updated every 5 s to capture relevant dynamics while maintaining performance.

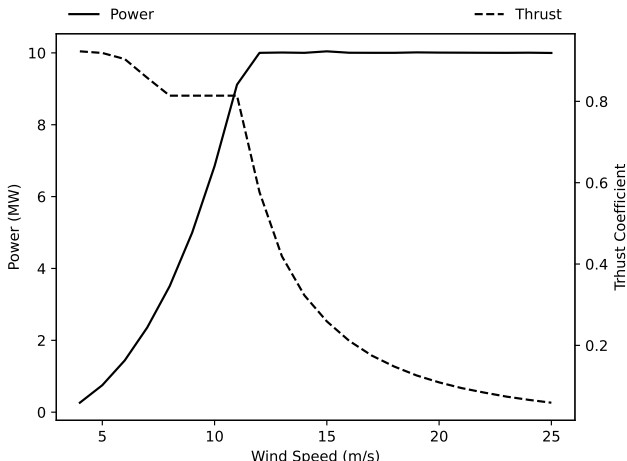

**Figure 1.** The power and thrust-coefficient curves of the reference DTU 10 MW turbine, based on the data in Bak et al. (2013b)

**Main bearing load estimation**: Hub loads calculated using HAWC2 were inputted to a simplified analytical drivetrain model which consists of two main bearings and a main shaft (approximated as rigid), shown in Fig. 2. This quasi-static model, previously applied by Hart et al. (2022), estimates main bearing loads via a static force and moment balance at each time step under the following assumptions:

1. All non-torque loads are reacted by the main bearings

2. The main bearings provide force response only, i.e. they do not support moments individually

3. The rotor-side main bearing carries the axial load

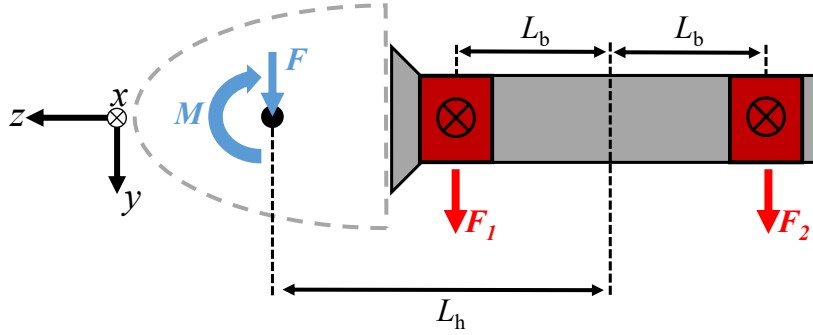

**Figure 2.** Force diagram of a main shaft in a four-point support drivetrain. Note, the $z$-axis is aligned with the turbine drivetrain and hence sits at a tilt angle relative to the ground. Force and moment vector components (including the definition of "positive direction") at each location are defined via the depicted HAWC2 reference frame.

4. The contribution of main shaft and/or gearbox or generator weight to main bearing loads is small and may be neglected for the purposes of this "relative impacts" study

5. Similarly, that the contribution of rotor and drivetrain weight to the axial load (via rotor-tilt) is small.

The reference frame for hub loads in HAWC2 is also provided in Fig. 2 and, importantly, differs from that used in (Hart et al., 2022). The applied model has $L_b = 1$ m and $L_h = 3.7$ m. The main bearing force response equations (also in the depicted HAWC2 reference frame) are:

$$F_1^x = -\frac{1}{2}\left(\left(1+\frac{L_h}{L_b}\right)F_h^x + \frac{M_h^y}{L_b}\right) \tag{4}$$

$$F_1^y = -\frac{1}{2}\left(\left(1+\frac{L_h}{L_b}\right)F_h^y - \frac{M_h^x}{L_b}\right) \tag{5}$$

$$F_2^x = -\frac{1}{2}\left(\left(1-\frac{L_h}{L_b}\right)F_h^x - \frac{M_h^y}{L_b}\right) \tag{6}$$

$$F_2^y = -\frac{1}{2}\left(\left(1-\frac{L_h}{L_b}\right)F_h^y + \frac{M_h^x}{L_b}\right) \tag{7}$$

$$F_1^z = -F_h^z \tag{8}$$

$$F_2^z = 0, \tag{9}$$

where $F_h^{x,y,z}$ and $M_h^{x,y}$ denote the axial, $z$, and radial, $x$ and $y$, hub-centre force and bending moment components obtained from HAWC2 models.

**Main bearing rating life assessment**: The modelled wind turbine drivetrains consist of a rotor-side double-row tapered roller bearing which supports both radial and axial loads, and a generator-side cylindrical roller bearing which supports radial loads only. The relevant parameters from ISO 281 for both bearings are given in Table 1. Bearing pitch diameters[3] are included, to

---

[3]This being the diameter of the circle connecting all rolling element centres around the bearing when each has the same amount of clearance to both races.

indicate the physical size of these bearings. For the locating (rotor side) bearing in this case, ISO 281 specifies $X = 1$ and $Y = 0.45 \cot(22.1°) = 1.108$ when $F_a/F_r \leq 1.5 \tan(22.1°) = 0.609$; and $X = 0.67$ and $Y = 0.67 \cot(22.1°) = 1.65$ when $F_a/F_r \geq= 0.609$. For the non-locating bearing $X = 1$ and $Y = 0$ at all times. These bearings were designed by the Offshore Renewable Energy Catapult, within a commercial project, for application in benchmarking and comparative analyses. The utilised bearing specifications should, therefore, be considered representative rather than definitive. Importantly, the current study is mainly concerned with *relative* effects and (from Equations 1 and 3) it can be shown that,

$$L_{\text{res}} \propto C_{\text{D}}^{10/3}. \tag{10}$$

Relative rating life impacts are, therefore, unchanged by the specific choice of $C_{\text{D}}$. Estimated radial loads at each bearing were combined into a radial load magnitude,

$$F_{r,i} = \sqrt{(F_i^x)^2 + (F_i^y)^2} \text{ for } i = 1 \text{ and } 2,$$

and, together with the estimated axial loads $F_{a,i} = |F_i^z|$, were inputted to ISO 281 basic rating life formulations (see Sect. 2.1). A rating life (in units of years) is calculated for each time step in each individual simulation, and a single simulation resultant rating life is calculated using Eq. 3, with equal weightings across all time steps (since the conditions present within each individual time step persist for the same amount of time).

**Table 1.** Bearing properties.

| Bearing | Nominal contact angle (deg.) | Dynamic capacity, $C_{\text{D}}$ (MN) | Pitch diameter (m) |
|---|---|---|---|
| Rotor side | 22.1 | 56.58 | 3.39 |
| Generator side | 0 | 46.21 | 2.91 |

## 3.2 Two-turbine parametric analysis

The above tool chain was applied to a simple two-turbine setup to investigate rating life impacts from single wakes. This parametric analysis was performed by placing the two turbines at a downstream distance of $5D$ from each other and then varying the cross-flow distance between the turbines (see Fig. 3a). We chose 5 rotor diameters as a quantity that is often examined (e.g. in Simley et al. (2020a)) and similar geometries have been used when validating novel control strategies (e.g. Simley et al. (2020b)). We also conducted the same analysis at 3D and 4D separations and found qualitatively similar results. The relevant inflow generation parameters are listed in Table 2. The mean inflow direction was kept constant (from left to right in Fig. 3a) with both turbines yawed so as to face into the wind. The analysis included a total of 31 cross-flow offsets (ranging from -1.5$D$ to 1.5$D$). Full sets of simulations were performed at hub-height mean wind speeds of 7.5, 11 and 15 m/s, which were chosen as operating points in the maximum efficiency, near-rated, and above-rated operating regions. Furthermore, the complete analysis was completed six times, using different turbulence seeds in each case. These simulations were each run for 2,000 s, discarding the first 1,000 s of results before processing. This was done out of an abundance of caution—the discarded

initial 1,000 s allows for the flow to fully develop within the domain and the remaining 1,000 seconds is nearly double the standard 600 s measurement period. As the Mann turbulence box is statistically stationary (Mann, 1998), there is not a danger in discarding the initial transient flow. The resultant rating lives were calculated for each set of input parameters and each turbulent seed individually, without further combination. The results of this analysis therefore give resultant rating lives which assume a given set of conditions hold across the full bearing lifetime. A total of 558 simulations were performed.

**Table 2.** Inflow Generation and Site Condition Parameters for the Wind Farm Simulation.

| Category | Parameter | Value | Description |
|----------|-----------|-------|-------------|
| **Turbulence** | Turbulence Model | Mann Model | The specific spectral model used to generate the stochastic, 3D ambient turbulence field. |
| | Turbulence Intensity (TI) | 5% | A low-to-moderate level of turbulence specified for the ambient wind field. |
| | Eddy Lifetime ($\Gamma$) | 3.9 | A dimensionless parameter defining the anisotropy of the turbulence due to wind shear. |
| | Length Scale (L) | 33.6 m | The characteristic size of the energy-containing eddies in the turbulent flow. |
| | Ambient Field Resolution | (3.0, 3.0, 3.0) m | The grid spacing (dx, dy, dz) of the main ambient turbulence box. |
| | Wake Turbulence Resolution | (3.8, 1.5, 1.5) m | The grid spacing (dx, dy, dz) of the turbulence box for each turbine wake. |
| | Turbulent seeds | 6 | Number of repetitions of each simulation (with given parameter values) using different turbulent seeds |
| **Mean Wind** | Shear Model | Power Law | The model describing the increase of mean wind speed with height above the ground. |
| | Shear Exponent ($\alpha$) | 0.2 | The exponent defining the shape and steepness of the vertical wind shear profile. |
| | Wind speeds (two turbine analysis) | 7.5, 11, 11.5 m/s | Hub-height mean wind speeds used in two-turbine simulations |
| | Wind speeds (wind farm analysis) | 6, 8, 10, …, 24 m/s | Hub-height mean wind speeds used in wind farm simulations |
| | Inflow direction (wind farm analysis) | 0, 5, 10, …, 355° | Mean direction of the flow passing through the simulated wind farm |

## 3.3 Wind farm analysis

The same tools were applied to undertake a complete wake-inclusive full-wind-rose (i.e. all wind directions) main bearing rating life assessment for all main bearings within the 32 turbine TotalControl reference wind farm (Andersen et al., 2018). This wind farm has an east-west turbine spacing of $10D$ and a north-south spacing of $5D$, where $D$ is the rotor diameter. Since the site is conceived of as being offshore, there are no modelled terrain effects. The site is assumed to have the same Weibull wind speed distribution for each inflow direction, with $k = 2$ and an annual mean wind speed of 10 m/s, with the same inflow generation parameters listed in Table 2. Simulations were undertaken for a total of 72 inflow directions (5° direction bins). For each inflow direction, simulations were performed for hub-height ambient mean wind speeds from 6 m/s to 24 m/s, in 2 m/s increments. A power-law vertical wind shear profile was applied throughout, with shear exponent 0.2. These large wind farm cases were run for 2,000 s with the first 1,000 s being discarded, except for the cases with 6 m/s and 8 m/s inflow, which were run for 3,000 s with the initial 2,000 s being discarded. This was again done out of an abundance of caution—these lower wind speed cases did not quite fully develop after 1,000 seconds, so 2,000 seconds are discarded, and the remaining 1,000 seconds are examined. Simulations using each set of parameters were repeated six times, using a different turbulent seed for each. A total of 4,320 simulations were performed, using DTU's Sophia supercomputer (Technical University of Denmark, 2019). A flow-field visualization of one such simulation, in which the wind farm layout can also be seen, is shown in Fig. 3b. For each main bearing within the wind farm and each wind direction: 1) outputs of each simulation were first combined

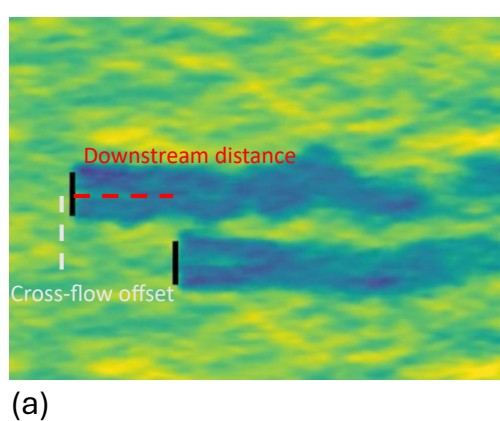

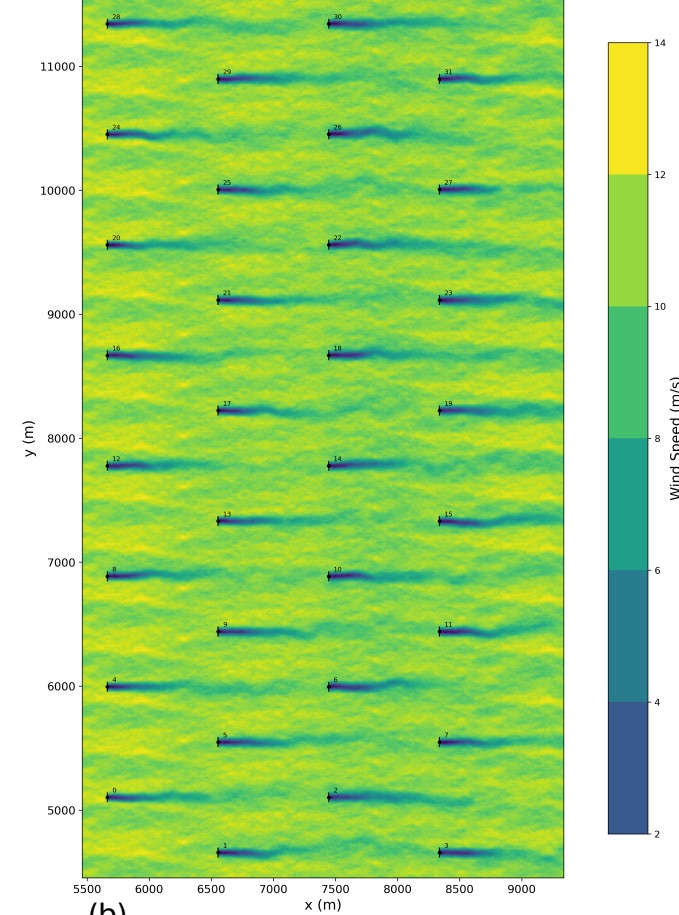

(a)

(b)

**Figure 3.** Layouts considered in this study. (a) A snapshot of the two-turbine parametric analysis. The downstream distance is held at $5D$, and the cross-flow offset distance between the turbines varied. Note, a negative cross-flow offset is shown in the figure. (b) Layout and flow-field visualization of the TotalControl 32-turbine reference wind farm (Andersen et al., 2018), including wake effects. Turbine indexes are also given.

into a single simulation rating life, as previously described; 2) turbulent seed results were then combined for each value of mean wind speed, again using equal weightings; 3) a rating life combination over wind speed values was then undertaken, with weightings determined (see Kenworthy et al. (2024)) using the specified Weibull distribution. At this stage, each bearing

therefore had a single resultant main bearing rating life associated with each inflow direction. A single, final, resultant rating life was then determined, for each bearing, using weightings specified by a generalized elliptical wind direction rose (Hart, 2025). Given the expected sensitivity of results to wind direction, a range of direction roses were considered. Based on model fitting

to offshore wind farm data in Hart (2025), sensible/realistic ranges for wind rose model parameters $a$ and $f$ were identified $(0.6 \leq a \leq 0.8,\ 0.1 \leq f \leq 0.6)$, and results were determined for a range of wind roses which span these values (see Fig. 4). The prevailing (highest probability) wind direction corresponded throughout to the left-to-right flow direction apparent in Fig. 3b.

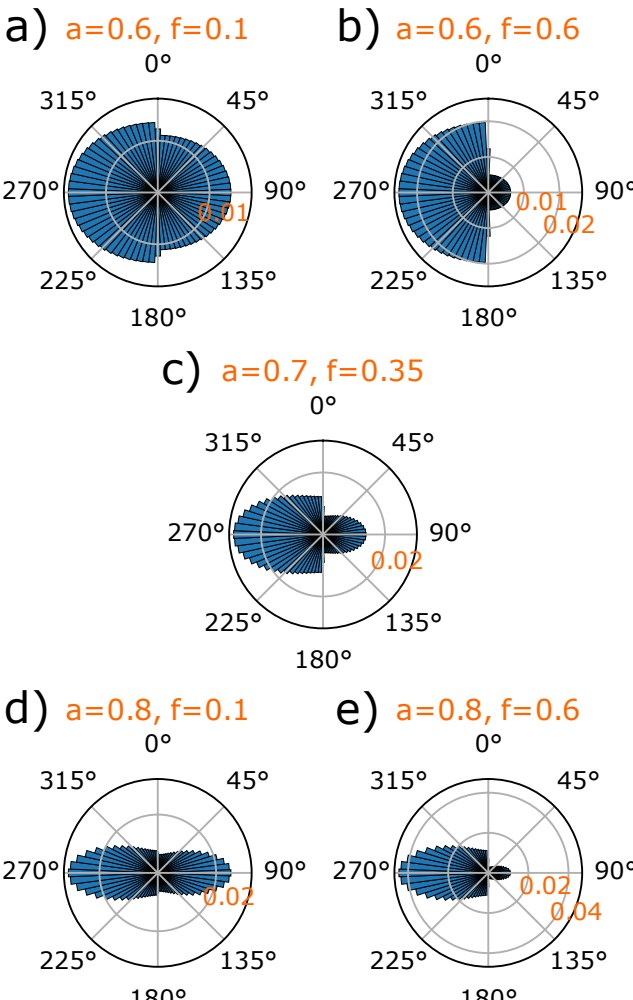

**Figure 4.** Edge case wind direction roses, in terms of $a$ and $f$ values, are shown in subplots a, b, d and e. The full envelope of applied wind direction roses fell between these bounding cases, each with a prevailing wind direction of $270°$. The "median" wind rose, *i.e.* with $a$ and $f$ values at the centre of each variable's range, is shown in subplot c.

## 4 Results

Results are presented in the current section for both the two-turbine parametric analysis and the wind farm study. Given the main bearing which supports both axial and radial loads (the rotor-side bearing in the current paper) is known to be the most common to fail (Hart et al., 2023), that bearing will be the main focus here.

### 4.1 Two-turbine parametric analysis results

The results of the two-turbine parametric analysis are presented in Fig. 5. Considering the (unwaked) front turbine results first, rotor- and generator-side bearing rating lives can be seen to far exceed the minimum design life of 20 years, as would be expected based on previous work (Kenworthy et al., 2024). This again highlights the key domain-challenge presented by main bearings which have a sufficiently large dynamic capacity, $C_D$, according to rating life formulations (Kenworthy et al., 2024), but which still failure prematurely in large numbers (Hart et al., 2023; EPRI, 2024). As discussed in Kenworthy et al. (2024), interpretation of such rating lives is nuanced, since rating life formulations are generalized, simplified and do not account for all mechanisms of failure. The current analysis is, however, principally interested in the relative impacts of wake impingement in this setting as a route to considering questions of bearing load modelling sufficiency. Given the rating life results obtained for the back (waked) turbine, significant wake-driven impacts are observable. Interestingly, this includes a strong asymmetry, depending on the direction in which an offset occurs (recall that a negative offset is that shown in Fig. 3a). The observable asymmetry also becomes more pronounced at higher wind speeds, possibly as a result of increased aerodynamic sensitivities stemming from wind-speed-squared lift and drag terms. Differences are also observable between wake impacts seen for the rotor-side versus generator-side main bearing, due to the former reacting axial and radial loads and the latter reacting radial loads only.

As hypothesised in Kenworthy et al. (2024), wake impingement can reduce the main bearing rating life, with negative offset partial impingement (-0.5$D$) driving reductions of as much as 73.5 % for the rotor-side bearing and 96.7 % for the generator-side bearing. Despite the larger percentage reduction on the generator side, the smallest rating life is still observed for the rotor-side main bearing (116 years on the rotor side, versus 212 years on the generator side). Positive offsets, on the other hand, can either elicit a less dramatic reduction in rating life or increase rating life above that of the unwaked case (as seen for 15 m/s wind speeds). Subsequent investigations of this phenomenon revealed these effects are the result of wake-driven aerodynamic blade load perturbations and their interactions with rotor weight. To elaborate, a positive offset partial-wake causes the downwind turbine to experience a reduced flow velocity across downward traveling blades. Lift for such blades acts towards the ground and so augments the gravitational force. Reducing the lift on such blades therefore lessens the augmentation effect and so reduces the mean load on the hub and therefore the main bearing. This in turn causes an increase in the bearing rating life. Alternatively, a negative offset partial-wake causes the downwind turbine to experience a reduced flow velocity across upward traveling blades. Lift for such blades acts away from the ground and against the gravitational force. Reducing the lift on such blades therefore lessens the countering effect and increases the mean load on the hub and therefore the main bearing. This in turn causes a reduction in the bearing rating life. Such interactions with gravity as the principal driver of

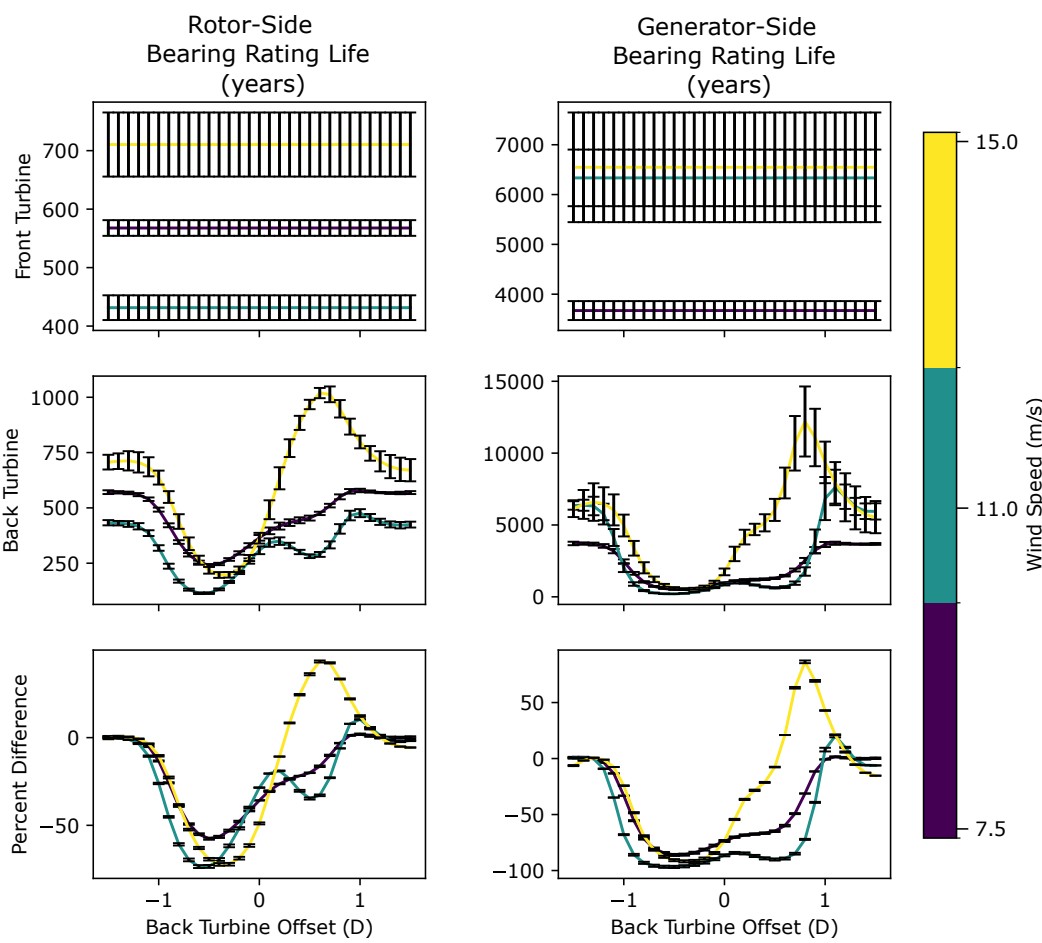

**Figure 5.** Results of the two-turbine parametric analysis. Error bars indicate seed-to-seed variation across six turbulent seeds.

observed asymmetries is clearly demonstrated by undertaking an identical analysis in the absence of gravity. As shown in Fig. A1 (see Appendix A), removing gravitational forcing removes the observed asymmetries. These results highlight the fact that, due to gravitational loading, the baseline main bearing mean load is significantly offset from zero. Subsequent effects and aerodynamic interactions therefore need to be considered relative to that nonzero starting point. Overall, the two-turbine parametric analysis results indicate that main bearing rating lives are strongly influenced by wake impingement (on the order of 75 % for the axially supporting bearing), with the most significant reductions seen for partial wake impingement ($-0.5D$) occurring across upward traveling blades. Within a wind farm, the standard grid spacing between turbines will commonly be on the order of $6D$–$10D$ in the prevailing wind direction, and $3D$–$5D$ in the cross-wind direction (Manwell et al., 2010). As the wind direction changes the cross-flow offset between turbines (relative to the inflow direction) will vary continuously across the full range analysed in this parametric analysis and beyond.

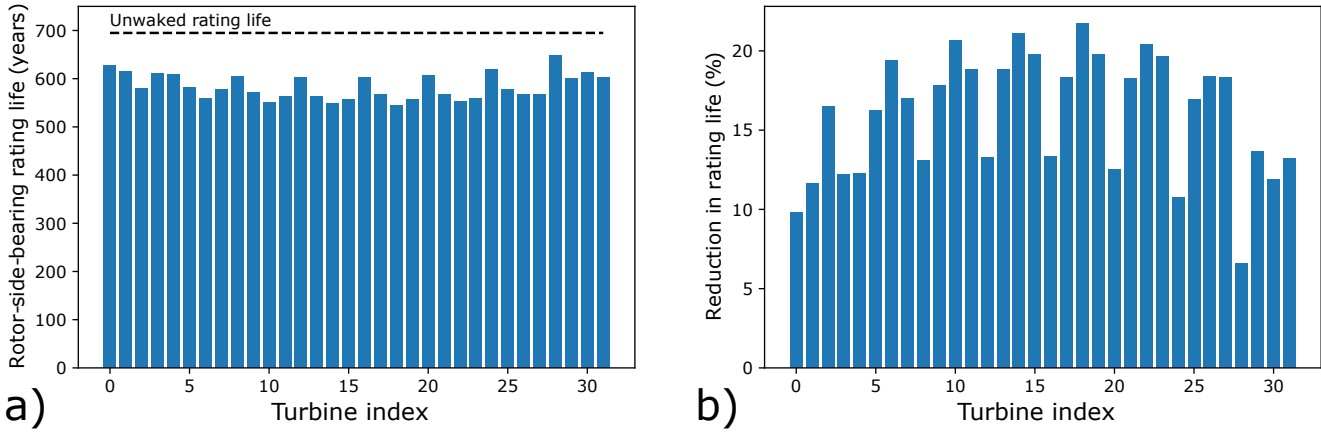

**Figure 6.** a) Rating life results for all rotor-side main bearings, obtained from a wind direction rose with parameters $a = 0.7$ and $f = 0.35$. The rating life corresponding to unwaked operation is also shown. b) Percentage reductions in rotor-side main bearing rating lives, for the same wind rose, relative to the unwaked rating life.

## 4.2 Wind farm results

Wind farm analysis results for the axial load support rotor-side main bearing are now considered. As a baseline, the unwaked bearing rating life was determined for all bearings in the left-hand row of Fig. 3b, using all turbulence seeds and wind speeds while keeping the inflow direction fixed (left-to-right only) to ensure no wake impingement occurred. These front-row rating lives were then averaged to obtain a single representative value for the unwaked rating life. Figure 6a presents rating life results for all turbines for the wind rose with parameters taken at the centre of each parameter range, $a = 0.7$ and $f = 0.35$ (see Fig. 4c). The baseline unwaked rating life is also shown. Figure 6b shows the same results, expressed as the percentage reduction in main bearing rating life (due to wake impingement) relative to the unwaked case. First, it is important to note that a decrease in rotor-side main bearing rating life was obtained for every turbine in the wind farm. Therefore, while it has been shown that some specific conditions can result in rating life increases (see Fig. 5), the more commonly observed life-reducing cases appear to dominate overall. The relative rating life reductions in Fig. 6b range from 6–22 %, with an average value of 16 %. A second key takeaway is that when accounting for a representative wind direction distribution, rating life reductions were found to be smaller than the worst-case scenarios observed during the parametric analysis.

It also proves instructive to consider which turbines experience the greatest wake-driven rating life reductions. The four greatest reductions occurred, in descending order, for turbines 18, 14, 10, and 22, all of which (see Fig. 6b) are located on the wind farm interior and within the "back" interior row relative to the prevailing wind inflow direction (left-to-right in Fig. 3b). The next four greatest reductions occurred for turbines 19, 15, 23, and 6. These turbines include one in the interior back row, and the remainder in the exterior back row, relative to the prevailing wind direction. Via the folding parameter, the wind rose has therefore imposed on the wind farm a notion of "front" and "back" turbines with the most severe rating life impacts affecting the more consistently waked turbines, these being in the back rows (interior and then exterior). Additionally, these

most impacted eight turbines are all centrally located with respect to the vertical ($y$-coordinate) layout of the farm; this is

another factor leading to a greater incidence of wake-impinged operation for these turbines.

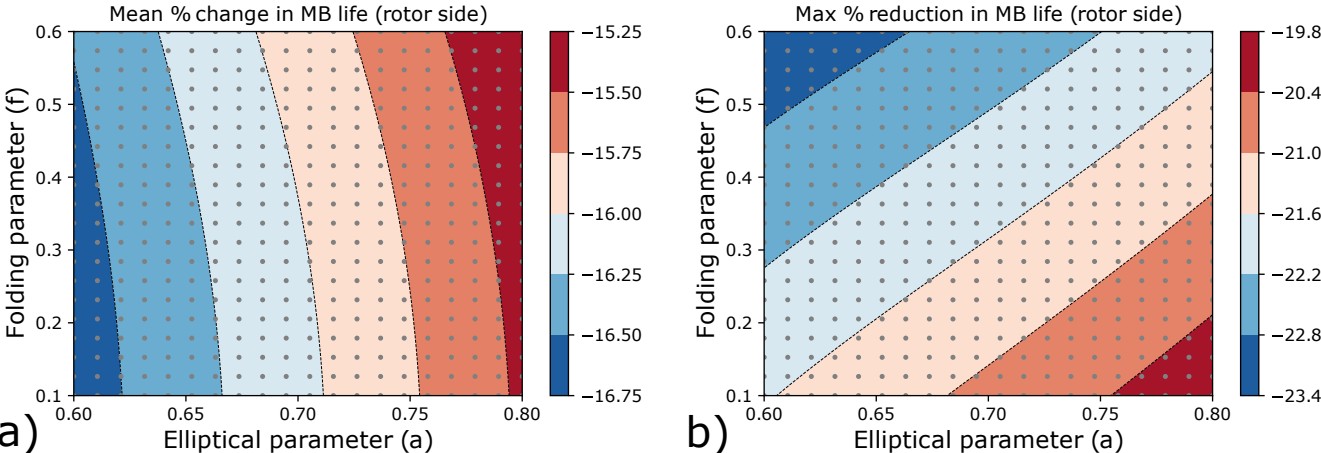

**Figure 7.** a) Mean percentage change in rotor-side main bearing rating lives (relative to the unwaked case) across the wind farm, as a function of wind direction rose parameters. b) Maximum percentage reduction in rotor-side main bearing rating life (relative to the unwaked case) observed within the wind farm, as a function of wind direction rose parameters. The grey dots indicate the parameter combinations considered, and therefore the data locations used to determine the resulting contours.

Wind farm rating lives were then determined for the full range of wind rose parameters. Figure 7 shows the results of this analysis, quantified via the mean percentage change in main bearing rating life across the wind farm (Fig. 7a) and the maximum percentage rating life reduction across the wind farm (Fig. 7b), both calculated relative to the unwaked case. Mean effects

can be seen to vary little as the wind rose parameters are adjusted; the observable changes are almost entirely driven by the elliptical parameter. This result makes intuitive sense, since the folding parameter symmetrically adjusts the wind rose direction dominance. As previously described, a notion of "front" and "back" is thus being imposed on the turbines; the low sensitivity of the mean value to changes in $f$ indicates that the (relative) rating life reductions at some locations are commensurate with the rating life increases in others. The impacts from varying $a$ are also intuitive, given that turbine spacings are smaller in the

vertical direction than in the horizontal (see Fig. 3b). This being the case, "rounder" wind roses mean more time spent with the wind directions in the vertical direction (the $y$-direction in Fig. 3b), where wake effects are stronger due to the smaller vertical spacing. Despite being small in magnitude, wind rose shape impacts are clearly discernible here. A different pattern emerges when considering the maximum reductions in main bearing rating lives across the wind farm (Fig. 7b). In this case we see a greater sensitivity to wind rose parameters, with maximum percentage reductions falling between 19.8 % and 23.4 %. Similar

logic as above holds for $a$. For $f$, turbines in the more commonly waked back rows of the wind farm spend increasingly large proportions of time in more deleterious waked conditions as the value of $f$ increases. Again, wind rose shape impacts are clearly discernible here and, interestingly, the results are approximately equally sensitive to the value of elliptical and folding parameters.

## 5  Discussion

Overall, the presented results indicate that wind farm main bearing rating lives are reduced by the effects of wake impingement, resulting in rating life reductions on the order of 16 % on average, and on the order of 20–25 % at maximum (for the axially fixed main bearing). Consistent with an earlier study (Krathe et al., 2025), rating life impacts were found to be asymmetrically related to the side on which impingement occurs due to differing interactions between aerodynamic force perturbations and their orientations relative to gravity. If one has a choice, the "better" side to partially wake is the one on which blades are

traveling towards the ground – a result which could prove useful in the design of wake steering control algorithms. The same analysis indicated that mean bearing loading is a strong indicator for rating life impacts (as has previously been noted (Kenworthy et al., 2024; Krathe et al., 2024)), which could prove useful if seeking to implement simple heuristic bearing impact metrics in layout and/or control co-design codes, similar to Stanley et al. (2023). A two-turbine parametric analysis demonstrated that the greatest reductions in rating life occur when a wake covers half of the downstream turbine rotor area

($0.5D$ impingement), another result with the potential to inform wake steering approaches. In contrast to the results of Krathe et al. (2025), our wind farm wake impingement results showed a decrease in rating life (relative to the unwaked case) for every rotor-side main bearing in the wind farm. This discrepancy could, among other things, relate to differences in wake modeling methodologies or floating/fixed turbine load response. Further consideration of these differing results is therefore warranted. Our wind farm results also indicated that rating life impacts are sensitive to the shape of the site wind rose, with the most

reduced lives associated with more unidirectional wind roses, as well as those with larger probability mass in directions along which turbines are spaced more closely.

The above results must be interpreted with care and, critically, with careful consideration of the appropriate interpretation of the ISO 281 rating life, in particular regarding its limitations (see Sect. 1 and 2, and Kenworthy et al. (2024)). Similarly, one must also appreciate the limitations of the present study, in which simplified (to different extents) representations were utilized

for modeling the ambient wind field, wakes, wind turbines and drivetrains. Since all simulations were undertaken for a low-to-moderate level of turbulence (5% TI), it is possible that the magnitude of observed wake impacts would be reduced in higher turbulence conditions where wake recovery is faster. The present study was also undertaken using 10 MW wind turbine models; in future work it would therefore be beneficial to quantify how main bearing wake impacts manifest at different scales. With the above caveats in place, and in the context of Sect. 1 and 2 discussions on modeling sufficiency, the following high-level

observations and recommendations are made:

1. Wake impingement has been found to reduce ISO 281 main bearing rating lives by 16 % on average, and by as much as 20–25 % at maximum.

2. Wake effects should be considered a necessary effect for model inclusion to help ensure the sufficiency of main bearing operational load models.

3. If other effects (such as drivetrain and bearing housing elasticity, inertial loading or those which may be accounted for using ISO 16281) are also incorporated, this could conceivably result in ISO-derived (281 or 16281) rating lives which

can account for the reported rates of main bearing failures. This, in turn, would be evidence in support of a view that rolling contact fatigue might in fact still underpin many (or even most) main bearing failures.

4. Future work should therefore look to establish a realistic and detailed high-fidelity modelling chain that both captures and combines the various important rating life reduction effects, which now include wake impingement, while remaining open to the possibility that alternative failure modes (not captured within rating life formulations) may still underlie a majority of main bearing failures. Validations against field data would be an important consideration during such future work.

## 6 Conclusions

This paper sought to quantify the relative change in main bearing rating lives across a wind farm when including the effects of wake-impinged operation. Beyond evaluating the extent to which wakes may be a contributing factor in premature main bearing failures, this research was also motivated by questions concerning the sufficiency of models used to assess main bearing design and reliability. Background context and modelling techniques were discussed, and a modelling tool chain was established for quantifying the effects of wake impingement on main bearing rating lives across a wind farm populated by 10 MW wind turbines. Turbine loads were computed using the Dynamiks Python package, including application of a dynamic wake meandering model. A two-turbine parametric analysis was undertaken first, showing that partial wake impingement can reduce the axially supporting main-bearing rating life associated with a single set of conditions by as much as 50–75 %. These parametric results do not account for varying site conditions. Rating life impacts were also found to be asymmetrically related to the side on which impingement occurs, indicating that, for the main bearing, there may be a "better" side for wake impingement to occur. A full wind farm analysis was then undertaken for the TotalControl 32-turbine reference wind farm, including full wind rose simulations across all operational wind speeds. The full wind farm analysis results properly accounted for varying site conditions via a Weibull wind speed distribution and a range of parametric wind direction rose models. Results showed that, overall, the wind farm main bearing rating lives were reduced by the effects of wake impingement on the order of 16 % on average and as much as 20-25 %, both for the locating main bearing. Sensitivities of the maximum rating life reductions to the shape of the site wind rose were also discernible. It was then highlighted that these results must be interpreted with careful consideration for methodological limitations, an appropriate interpretation of the ISO 281 rating life, and with an appreciation for the modelling simplifications which are present in the overall tool chain. Based on the outlined research findings, it was concluded that wind farm-level wake effects should necessarily be included when undertaking main bearing operational load modelling and rating life assessment. It was also recommended that a realistic and detailed high-fidelity modelling chain be developed in future work (and validated against field data) to combine various important rating life reduction effects alongside wake impingement.

**Acknowledgments**

Edward Hart is funded by an EPSRC Innovation Launchpad Network+ Researcher in Residence Fellowship (EP/W037009/1), a collaboration with the Offshore Renewable Energy Catapult. This research has been supported by the SUDOCO project, which is funded through the European Union's Horizon Europe Programme under grant agreement No. 101122256. Numerical simulations were run on the Technical University of Denmark's Sophia supercomputer (Technical University of Denmark, 2019). This work was also authored in part by the National Laboratory of the Rockies operated by the Alliance for Sustainable Energy, LLC, for the U.S. Department of Energy (DOE) under contract no. DEAC36-08GO28308. Funding was provided by the U.S. Department of Energy Office of Energy Efficiency and Renewable Energy Wind Energy Technologies Office. The views expressed in the article do not necessarily represent the views of the DOE or the U.S. Government. The U.S. Government retains and the publisher, by accepting the article for publication, acknowledges that the U.S. Government retains a nonexclusive, paid-up, irrevocable, worldwide license to publish or reproduce the published form of this work, or allow others to do so, for U.S. Government purposes.

*Data availability.* The scripts used to conduct this analysis are available here: https://doi.org/10.5281/zenodo.15148660

*Author contributions.* J.Q., E.H., and Y.G. conceived the study. J.Q., E.H., M.B.N., R.S.L., and J.L. implemented the computational framework. E.H. developed the bearing life analysis. All authors contributed to the interpretation of results and writing of the manuscript.

*Competing interests.* Some authors are members of the editorial board of Wind Energy Science.

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

**Appendix A**

The two-turbine parametric analysis was repeated without a simulated gravity force. These results are shown in Fig. A1 and
515 indicate that the previously observed asymmetries are predominantly due to interactions between aerodynamic and gravitational loads.

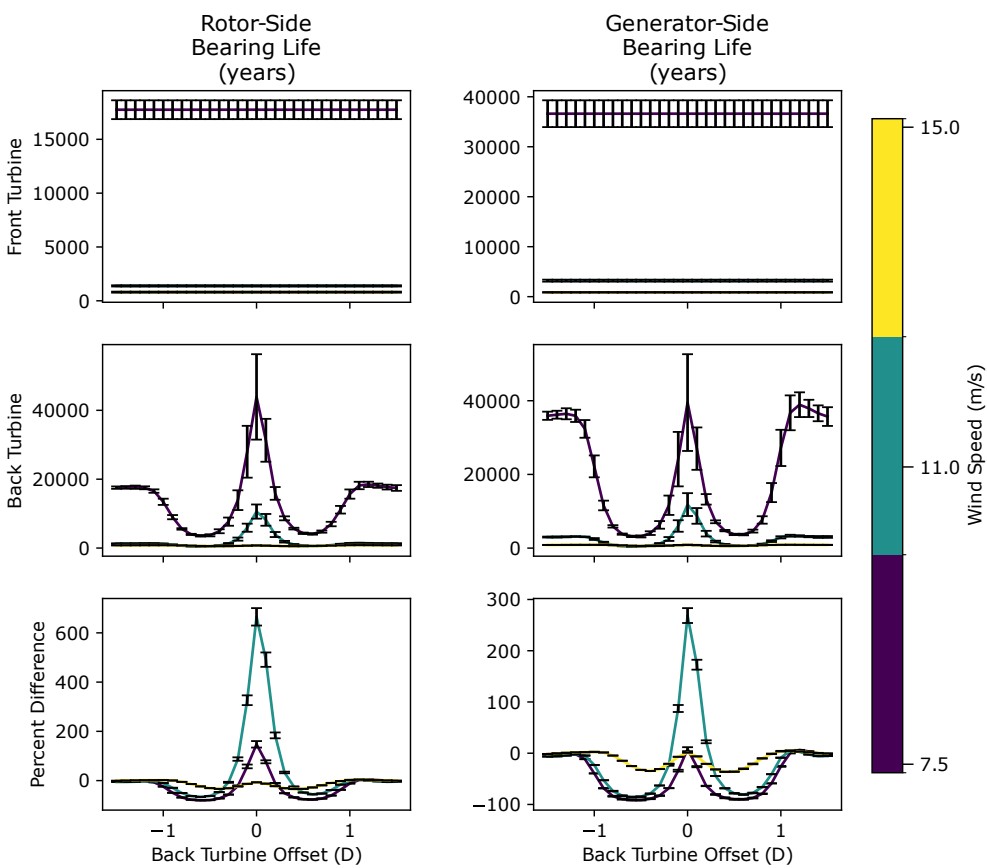

**Figure A1.** Results of the two-turbine parametric analysis with gravity turned off. Error bars indicate seed-to-seed variation across six turbulent seeds.