# Peer review of "Reductions in wind farm main bearing rating lives resulting from wake impingement"

_Wind Energy Science, 2025_

## Referee Comment (RC1)

**Review**

**General comment:**
This manuscript presents a simulation-based study on the impact of wake impingement on the main bearing basic rating life across a wind farm using DTU 10 MW turbines. By integrating wind field modeling, turbine loading, and fatigue life assessment, the authors aim to quantify how partial wake conditions reduce main bearing life, both in isolated two-turbine cases and a full wind farm layout. The topic is timely and relevant, addressing an underexplored but important contributor to drivetrain failures in large-scale wind turbines.
The paper is generally well-structured and technically ambitious. It provides useful insight into how wake conditions can lead to significant reductions in basic rating life, and how wind rose shape and wake offset can amplify or mitigate this effect. However, there are a number of technical and methodological concerns that must be addressed. These include inaccuracies in the use of ISO 281 terminology, oversimplified drivetrain assumptions, unclear justifications for input parameters, and inconsistencies in geometries. The comments are as follows:

1. It seems there is a conflict between line 10 and line 19. Is there something missing in your calculation that is not consistent with the literature (Hart et al., 2023; EPRI, 2024) or just simplification in the calculation of the life? Please elaborate on how to correct the sentence in line 10 to resolve this conflict.

2. Line 53 defines rating life as the life that 90 % of the bearing population is expected to attain or exceed. It is not a completely correct definition, and it seems the authors mean basic rating life, $L_{10}$ (ISO 281), by this definition. Keep in mind that with a different $a_1$, life modification factor for reliability, instead of value 1, different reliability and life will be achieved (Table 12 from ISO 281 standard).

3. In formula 1, the basic dynamic load rating is defined as $C_D$. Because the formula is intended to be for a radial roller bearing (using 10/3 for p value in L10 formula), it is recommended to stick to the ISO symbols and use $C_r$ instead, which is the basic dynamic radial load rating. I'm not sure if the notation $C_D$ is common in the industry. It is the same with the dynamic equivalent radial load, $P_{eq}$.

4. In equation 3 the life is defined in the form of different proportions of time spent. Although this definition is not wrong and is used in some references, the $L_{10}$ is originally defined per revolution instead of time spent. Even when you want to sum the different operating conditions, the summation should be done on different numbers of revolutions instead of the time spent. In addition, it is not clear how the lives in revolutions unit changed to years units in the results.

5. In lines 73 to 78, the paper clarified why ISO 281 rating life is used. Strictly speaking, the paper used basic rating life because ISO 281 also proposes another, more advanced rating life named modified rating life, $L_{nm}$.

6. Line 140 presented the Weibull distribution as a standard model. Although it is true, it is important to say that considering the shape parameter of 2 leads to the Rayleigh distribution, which is a special state of the Weibull distribution, and it is presented in the IEC 61400-1 as well.

7. In the wind turbine simulation section, it is mentioned that the DTU 10 MW is considered; However, DTU 10 MW is a well-known reference wind turbine, it would be better to give some general information about its specification in this section, such as rotor diameter.

8. In Line 181, the axial dimension of the middle of the shaft is presented from the hub. The value is 3.7 m, and if the length between the center of the bearing and to center of the shaft is deducted, the remaining length is 2.7 m (between the center of the front bearing and the center of the hub). According to DTU 10 MW specifications, the hub diameter is 5.6 m, which leads to 2.8 m in radius. Even with a main shaft-hub connection diameter of 3 meters, the length between the center of the hub and the shaft-hub connection would be 2.36 m, and there is only 0.34 m for the distance between the center of the front bearing and the shaft-hub connection. This value is so unrealistic.

9. In line 191, it is presented that the weighing is equal across all time steps. Does the turbine rotate at a constant speed? Otherwise, how can such an assumption be justified? If this is not the case, please clarify what the assumption means.

10. The information about the bearings is limited to lines 186 to 193 and table 1. Please provide more details such as type of bearings, general dimensions, Value X and Y (dynamic radial and axial load factors), and manufacturer design code.

11. In line 194, 5D is considered for the downstream distance between turbines. Because this value will be fixed for the whole simulation in parametric analysis, it needs to be referenced and justified.

12. In line 198, three different wind speeds are assumed. None of them are rated or cut-out wind speeds. It is not clear to readers how these wind speeds were chosen.

13. In line 200, it is mentioned that the simulation was 2000 s and 1000 s is discarded. There are two notes in this item. First, what is the reason for doing a 2000-second simulation? To my understanding standard proposes a 10-minute simulation. Second, 1000 s discarding means putting away half of the data. If one discards half of the data, do the turbulence intensity and the characteristics of the wind remain untouched? Please ensure wind field characteristics remain consistent post-discard.

14. In section 3.2, it is not clear what the wind characteristics are. Please include a table describing wind and site conditions.

15. In section 3.3, the questions about simulation time (comment 13) are valid. The questions are more significant for 6 and 8 m/s.

16. In section 3.3, it is not clear what wind characteristics are besides an annual mean wind speed.

17. In Fig. 2b, a matrix of 32 turbines is considered in a symmetric pattern. Because of a symmetric pattern, even with consideration of different flow angles, it seems 16 turbines are enough. Please explain more about using 32 wind turbines and justify this assumption.

18. In Fig. 2b, there is no information about the arrangement and distances of the turbines. Also, out of curiosity, is there any reason to start turbine numbers from 0? It is suggested to add a table of turbine coordinates and naming conventions to make the layout traceable.

19. In line 215, equal weighing is used in rating life for turbulent wind model conditions. How do the authors justify such an assumption by considering variability in shaft speed due to the turbulence regime?

20. According to Fig. 4, the life results for the bearings for both turbines are in a different order. How do the authors describe such a phenomenon? Is the assumption of the taking loads by the front bearing reasonable? It should be noted that the bearings have the same order of radial dynamic load rating, and when the life has a different order, it shows a different order of loads.

21. Interestingly, the trend of the basic rating life of the bearing in the rotor and generator side regarding the wind speed in front turbine is different. The life of the rotor side at 7.5 m/s is higher than 11 m/s. On the other hand, on the generator side, the bearing life at 7.5 m/s is almost half of the other wind speeds. It would be valuable if the authors discussed more about this happening.
22. The results in Fig. 4 show the difference between the life of the bearings in front and back turbines. It would be useful to present the average power in the same figure, as maybe less power is one of the reasons behind the shorter life of the bearings.
23. In line 257, the standard grid spacing claim needs a reference.
24. Please add the illustration of the wind rose used in Fig. 5a. It can be added in Fig.3.
25. In line 267, it is observed that the basic rating life always decreases in the bearing, while in a few specific conditions in the parametric study, the life increases. To have a fair claim, the condition of the turbine in spacing and wind conditions should be the same.
26. Typing error in line 37: Redundant Kenworthy et al.

---

## Referee Comment (RC2)

**Summary:**

This study evaluates the impact of wake effects on main bearing rating lives considering 10-MW landbased wind turbines with four-point supported drivetrains and two main bearings, of which the upstream bearing carries the axial loads. Two studies are conducted. One considering two turbines separated in the mean wind direction by 5 rotor diameters, and with a variation of lateral positions of the downstream turbine. The second considers a 32-turbine wind farm subjected to environmental conditions supposedly representing a full operational life of the turbine. A variation of wind roses is considered.

This is an interesting, relevant and comprehensive study, with useful results. The text is well-formulated, and generally well organized. Some details are missing, and some assumptions related to environmental conditions can influence the conclusions significantly and should therefore be discussed. The following comments should be addressed:

**Introduction**

- Lines 35-43: Two questions are stated one related to validity of ISO-based main bearing rating life, and one related to what constitutes a realistic system model. I assume that this research is an attempt to answer the latter, but this is not very clearly stated. Am I correct? If not, the second question seems redundant. Please rephrase.
- Line 37: Duplicate reference to Kenworthy et al.
- Lines 44-46: Inconsistent use of "Sect." and "Section" please check the guidelines of WES.

**Background**

• The paper "Main bearing response in a waked 15-MW floating wind turbine in below-rated conditions" by Krathe et al looked at partial wake impingement effects on main bearing rating lives and should be referenced here. https://link.springer.com/article/10.1007/s10010-025-00808-z

**Section 2.1**

• Line 59: Radial and axial bearing loads are referred to here but not defined until p. 7. Please check give a brief description of them here.

**Section 2.2**

• Lines 109-110: The fatigue damage of the bearings depend highly on  $C_D$ , and it is not useful to compare the damage of the upwind and downwind main bearing without commenting on the difference in  $C_D$ .

**Section 2.3**

- Line 131: The reference applied for the Dynamiks Python package looks strange. Please check that it is presented as intended.
- Line 144: For someone not familiar with the model proposed by Hart, it is not trivial to understand what the elliptical and folding parameters describe. Please provide a brief explanation indicating what physical properties these parameters describe. In general, a more detailed description of this method would be useful to understand the results of this work.

**Section 3.1**

- Please provide more details related to the turbine. A table summarizing rated wind speed, hub-height, shaft tilt and rotor diameter would be useful. Is this a geared drivetrain?
- I assume that the wind farm is landbased (not offshore), but this is not stated anywhere. Please clarify.
- What is the rationale behind the choice of 5 % turbulence intensity? Turbulence intensity will significantly influence the wake recovery, which could alter the conclusions of this work. For a landbased turbine, 5 % is quite low compared to values recommended in the standards. It is important to discuss the validity of this assumption. The paper seeks to explain premature failure in main bearings, mainly reported for landbased turbines. If turbulence intensity is generally higher than what applied in this work, so that the wake recovers more quickly, it might not be valid to conclude that farm effects contribute that much to reduced main bearing lives.
- How is shear modeled in this work? If the power-law is applied, what shear exponent is used? Wind shear is highly important for main bearing rating lives. Combined with the wake deficit, the shear profile will determine what the "final" shear that the downstream turbine experiences. I.e. low shear could result in a "reversed" shear profile in which the wake velocity deficit (which is typically deflected vertically due to shaft tilt) leads to reduced mean wind velocity with height. Please clarify and discuss.
- It is common in industry to use a generator-side locating (carrying axial loads) bearing. To be relevant for industry, I would recommend reversing the setup (I assume this does not require running Dynamiks simulations over again but is related to post-processing).
- Please state what X and Y (load factors) are applied for each bearing. This is useful to understand the importance of thrust versus radial loads in the fatigue calculations.

- What is the rationale behind the choices of  $C_D$ ? Are these values representative of 10 MW turbines? The authors later (Section 4) comment on the high rating lives, but these results highly depend on  $C_D$ .
- The authors investigate a 10 MW turbine, while main bearing failure reports mainly exist for smaller turbines. Could the authors comment on whether wake effects can be generalized across turbine sizes? Could wake effects be less important for smaller turbines, and therefore not have result in the same reduction in main bearing rating lives?

**Section 3.2**

• Why is 5D applied in the two-turbine parametric analysis?

**Section 3.3**

- It could be useful to put the parameters presented here (e.g. k, annual mean wind speed, mean wind speeds, inflow directions etc.) into a table for better overview.
- What is the spatial grid resolution in the wind farm simulations and turbulent wind fields?
- Line 214: Suggest rephrasing to: "For each main bearing and each direction ":
- The distances between turbines along x and y should be stated more clearly
- Figure 2a: Axes missing.
- Line 221: The reference to Hart should not be in parenthesis.
- Line 221: "Based on model fitting to data" what kind of data? For what location are these ranges of wind roses realistic? What are the criteria for realistic? Is this data site-specific?
- Figure 3: Do all the wind roses evaluated have the prevailing direction of 180 degrees?

**Section 4**

- It could be useful to split this section into subsections to have a better overview of the different results.
- Line 228: The authors assume that the locating main bearing fails most commonly. What is this assumption based on? Why not present results of the rear bearing too (e.g. in the appendix)?
- Line 230 and 240: "...bearing rating lives can be seen to far exceed the minimum design life..." Again, the rating lives are dependent on the value of CD. A more detailed description of the choice of CD should be given if these findings should be considered important.
- Figure 4: Asymmetries are more pronounced for higher wind speeds. Could the authors comment on the differences in results between wind speeds?

- Lines 245-255: I think this explanation of the asymmetry is a bit too simple. Gravity mainly acts in the in-plane-bending moment in the blades and less so in the out-of-plane bending moment, depending on the shaft tilt and curvature of the blades. Out-of-plane blade root bending moments are predominantly important for main bearing loads (relative to in-plane BM). Is gravity in the blades driving hub pitch and yaw moments? When removing gravity, as presented in Fig. A1, the shaft moment due to rotor weight vanishes, and the radial loads are significantly reduced. With regards to the locating main bearing, bearing rating lives are now likely governed by axial loads, so that any asymmetry trend would disappear among the axial loads. It would be interesting to see a closer investigation of this effect.
- Line 257-258: "Within a wind farm, the standard grid spacing between turbines will commonly be on the order of 3D-5D". Is this referring to spacing in the predominant cross-wind direction? I believe that larger distances are seen in the predominant wind direction. A reference would be useful.
- Lines 275-285: Again, it would be useful to explain the physical meaning of *a* and *f* before discussing their impact on main bearing rating lives.

**Conclusion**

• The impact of turbulence intensity and shear on the results should be discussed.

---

## Author Response (AR1)

wes-2025-63: Reductions in wind farm main bearing rating lives resulting from wake impingement

**Response to Reviewer 1**

We thank the reviewer for their comments, which we feel have helped improved the quality and clarity of this manuscript. Reviewer comments are listed below, followed by our responses in blue. Further comments have now been added in red, outlining the changes which have been made.

1. It seems there is a conflict between line 10 and line 19. Is there something missing in your calculation that is not consistent with the literature (Hart et al., 2023; EPRI, 2024) or just simplification in the calculation of the life? Please elaborate on how to correct the sentence in line 10 to resolve this conflict.

To clarify, line 10 concerns bearing rating life (the life derived at the design stage based on simulated loading and ISO bearing life equations) while line 19 concerns the observed field life (calculated from failure data on operational turbine fleets). The gap between rating life (calculated) and field (observed) is the focus of much ongoing research, with the current paper seeking to understand if some of that gap may be the results on simulation and model based life estimation lacking wake effects. When revising the manuscript we will consider if this distinction can be more clearly outlined to the reader.

We have clarified this point by editing lines: Line 1-"This paper studies the impacts of wake impingement on main bearing rating lives predicted during the wind turbine design stage", Line 11-"it is important to note that these resultant rating lives (i.e. the predicted lives)", Line 20-" As will be elaborated on below, there is therefore a significant gap between the main bearing (predicted) rating life and (observed) field life."

2. Line 53 defines rating life as the life that 90 % of the bearing population is expected to attain or exceed. It is not a completely correct definition, and it seems the authors mean basic rating life, L10 (ISO 281), by this definition. Keep in mind that with a different a1, life modification factor for reliability, instead of value 1, different reliability and life will be achieved (Table 12 from ISO 281 standard).

This sentence was constructed this way to try and help ensure the concept is understandable to a wide audience, without additional baggage. This is why we wrote that "rating life is generally the life that 90% of the bearing population is expected to attain or exceed". We of course agree that the modified rating life can extend this other levels of reliability, but this is not done for the main bearing (as far as we are aware). The inclusion of the word "generally" was inserted to indicate some level of simplification (while also avoiding a full unpacking of the ideas), but we will revise this part of the manuscript to try and improve clarity.

For the sake of clarity we've done as you suggest and referred to the basic rating life directly here. We have additionally highlighted that the modified rating life is also a part of ISO 281, and clarify why we stick to the basic rating life here "While both the modified rating life of ISO 281 and the enhanced formulations of ISO 16281 seek to account for more detailed factors… given the focus of the current study is quantifying the *relative* impacts of wake impingement, the ISO 281 basic rating life formulation was applied"

3. In formula 1, the basic dynamic load rating is defined as CD. Because the formula is intended to be for a radial roller bearing (using 10/3 for p value in L10 formula), it is recommended to stick to the ISO symbols and use Cr instead, which is the basic dynamic radial load rating. I'm not sure if the notation CD is common in the industry. It

is the same with the dynamic equivalent radial load, Peq.

We adopted the notation used here in an earlier paper (https://doi.org/10.1002/we.2883), based on some of the literature and theory unpacked there. Since the current work is a very direct follow-on from that paper, we feel it is best to maintain consistent notation between these two papers.

We have kept the original notation for the reasons outlined.

4. In equation 3 the life is defined in the form of different proportions of time spent. Although this definition is not wrong and is used in some references, the L10 is originally defined per revolution instead of time spent. Even when you want to sum the different operating conditions, the summation should be done on different numbers of revolutions instead of the time spent. In addition, it is not clear how the lives in revolutions unit changed to years units in the results.

Thank you for raising this point. You are indeed correct that we neglect to mention that L10 is first calculated in revs, and then changed to time via a conversion that includes turbine rotational speed, while all these details appear in the earlier paper cited at this point in the manuscript (https://doi.org/10.1002/we.2883), we agree that more details on this in the current manuscript would also be beneficial. Concerning the "summation" across different operating conditions, we disagree with your assertion that this should necessarily be done in units of revolutions. One may combine across different operational conditions using any units (time or revs), so long as this is done correctly. Given the turbine operates at different rotational speeds, and that the full life of the turbine is not simulated directly (we simulate across expected conditions and then extrapolate to the full life), we argue that obtaining a resultant bearing life is more straightforward if one first converts to units of time. This happens by 1) calculating L10 in revs for set conditions 2) converting to L10 in years using a conversion factor that includes rotational speed 3) combining across different conditions by weighting according to the proportion of time spent in each condition. Note that this conversion naturally accounts for varying rotational speeds, removing the need for that to be accounted for later on.

Note, the manuscript already contains the clarification "ISO equations give $L_{10}$ values in millions of revolutions; these are then readily converted to units of time using the rotational speed of the wind turbine low-speed shaft". Given the combination into a resultant rating life can (and we argue should) be done using L10 in units of years, there is not a requirement for any further conversion after the resultant life is calculated.

5. In lines 73 to 78, the paper clarified why ISO 281 rating life is used. Strictly speaking, the paper used basic rating life because ISO 281 also proposes another, more advanced rating life named modified rating life, Lnm.

Fair point, we'll clarify here that we're using the ISO 281 basic rating life.

This clarification has been added, thanks for highlighting this!

6. Line 140 presented the Weibull distribution as a standard model. Although it is true, it is important to say that considering the shape parameter of 2 leads to the Rayleigh distribution, which is a special state of the Weibull distribution, and it is presented in the IEC 61400-1 as well.

While this is true, we don't feel it's particularly important to point this out in the manuscript.

As indicated, we didn't feel this would add anything in particular to the paper.

7. In the wind turbine simulation section, it is mentioned that the DTU 10 MW is considered; However, DTU 10 MW is a well-known reference wind turbine, it would be better to give some general information about its specification in this section, such as rotor diameter.

Agreed, we will add some relevant information about the rotor diameter and rated wind speed etc, as well as a diagram of the power and thrust curves for the DTU turbine.

We have added the hub height, rotor diameter, and a figure of the power and thrust curves (Figure 1). Note that this figure is based on previous simulations, not the results of the simulations presented in the study.

8. In Line 181, the axial dimension of the middle of the shaft is presented from the hub. The value is 3.7 m, and if the length between the center of the bearing and to center of the shaft is deducted, the remaining length is 2.7 m (between the center of the front bearing and the center of the hub). According to DTU 10 MW specifications, the hub diameter is 5.6 m, which leads to 2.8 m in radius. Even with a main shaft-hub connection diameter of 3 meters, the length between the center of the hub and the shaft-hub connection would be 2.36 m, and there is only 0.34 m for the distance between the center of the front bearing and the shaft-hub connection. This value is so unrealistic.

This appears to be a simple misunderstanding. The "hub diameter" for the turbine in question concerns the hub's radial size in the rotational plane (the plane through which the blades sweep). This dimension is therefore orthogonal to that of the drivetrain. As the hub is not a sphere, this same dimension cannot assume to provide any information concerning the plane in which the drivetrain sits. The drivetrain dimensions used here are in line with those of other turbines of a similar size in the literature (e.g. https://doi.org/10.1002/we.2476) and are both sensible and consistent with the turbine model being used.

For the reasons outlined here, no changes were made to the manuscript in response to this comment.

9. In line 191, it is presented that the weighing is equal across all time steps. Does the turbine rotate at a constant speed? Otherwise, how can such an assumption be justified? If this is not the case, please clarify what the assumption means.

The benefit of working with basic L10 lives converted to units of time is that varying rotational speeds have already been accounted for, and it is simply the relative time spent under each condition that determines the resultant life ratings. In our prior paper on this topic (https://doi.org/10.1002/we.2883) we showed that resultant lives may be calculated in stages, rather than needing to combine across all cases and conditions in one go. At this stage of the paper we are therefore combing bearing lives associated with each time-step of a single simulation, into a single resultant bearing life for that simulation. In this instance, a bearing life associated with each time step in the simulation was calculated, and each of those persist for the same amount of time (one timestep). Hence, they all have equal weighting. Later in the analysis, these rating lives are further combined using Weibull distribution and wind rose weightings, so the equal-weighting comment only applies to this first step to go from lots of timesteps in a simulation to a single resultant bearing rating-life for that simulation. We will review out discussion of this in the manuscript when revising, to see if these points can be clarified.

Some clarifying additions have been added: Line 219-" A rating life (in units of years) is calculated for each time step in each individual simulation, and a single simulation resultant

rating life is calculated using Eq. 3, with equal weightings across all time steps (since the conditions present within each individual time step persist for the same amount of time)."

10. The information about the bearings is limited to lines 186 to 193 and table 1. Please provide more details such as type of bearings, general dimensions, Value X and Y (dynamic radial and axial load factors), and manufacturer design code.

We will seek to provide further details of the bearings themselves in the revised manuscript. Note, off the shelf bearings are not generally suitable for application in such a large wind turbine, and technical specifications for more bespoke commercial offerings are not publically available. The bearings analysed in the current work were therefore shared by project partners in the Offshore Renewable Energy Catapult, who developed a drivetrain design as part of a benchmarking study for a commercial project. As the detailed design and study results from that commercial project are proprietary, some details may not be shareable. We will however include what we can. We would also point out that the current analysis is mostly concerned with the "relative" impacts of waking, which can be shown to remain the same even if the bearing dynamic rating changes. We will elaborate on that point too when revising the manuscript, as it helps demonstrate a broader generalisability for our results.

More complete information concerning the modelled main bearings has been added. This includes that fact that the rotor-side bearing is a double row tapered roller bearing, and that the generator-side bearing is a cylindrical roller bearing. Their pitch diameters have also now been included in Table 1. Note X and Y values are determined using the bearing contact angle and axial to radial load ratio as codified in ISO 281. We now also include some contextual information on where the bearing design came from (as described above). Finally, we now also highlight to the reader that resultant rating lives are readily shown to be proportional to $C^{(10/3)}$ (for C the dynamic capacity). It follows from this that the relative rating life impacts, with which we're primarily concerned, are not effected by the specific choice of C.

11. In line 194, 5D is considered for the downstream distance between turbines. Because this value will be fixed for the whole simulation in parametric analysis, it needs to be referenced and justified.

We determined that 5D was a fair representation of typical distances between turbines. In our analysis, we did include 3D and 4D spacing. However, we felt the trends revealed in the 5D analysis were representative. In other words, the presence/location of a partial wake is much more influential than its exact magnitude. We will add further discussion of this when revising the paper, and a reference to justify 5D as a reasonable nominal value.

We have added the explanatory text: Line 225-"We chose 5 rotor diameters as a quantity that is often examined (e.g., in Simley et al. (2020a)) and similar geometries have been used when validating novel control strategies (e.g. Simley et al. (2020b)).

12. In line 198, three different wind speeds are assumed. None of them are rated or cut-out wind speeds. It is not clear to readers how these wind speeds were chosen.

These wind speeds were chosen to represent different regions of the turbine's power-curve. We will add this clarification into the revised manuscript.

Clarification added: Line 231-" which were chosen as operating points in the maximum efficiency, near-rated, and above-rated operating regions."

13. In line 200, it is mentioned that the simulation was 2000 s and 1000 s is discarded.

There are two notes in this item. First, what is the reason for doing a 2000-second simulation? To my understanding standard proposes a 10-minute simulation. Second, 1000 s discarding means putting away half of the data. If one discards half of the data, do the turbulence intensity and the characteristics of the wind remain untouched? Please ensure wind field characteristics remain consistent post-discard.

We will add to the text to explain that the 1,000 second simulation times were done to edge on the side of more data, as opposed to the standard 600 seconds that is a standard amount. The burn-in perdiods were selected to ensure that the flow simulation had developed for several flow passthroughs before recording the relevant measurements. Discarding burn-in period data ensures the targeted wind field characteristics are representative, by removing earlier transients which would diverge from those which are being sought. As such, wind characteristics are preserved by this process.

We have added more justification: Line 233- "This was done out of an abundance of caution—the discarded initial 1,000 s allows for the flow to fully develop within the domain and the remaining 1,000 seconds is nearly double the standard 600 s measurement period. As the Mann turbulence box is statistically stationary (Mann, 1998), there is not a danger in discarding the initial transient flow."

14. In section 3.2, it is not clear what the wind characteristics are. Please include a table describing wind and site conditions.

Thank you for this suggestion. We will include these characteristics as a table, and provide some additional explanation.

The relevant inflow generation parameters are now listed and described in Table 2.

15. In section 3.3, the questions about simulation time (comment 13) are valid. The questions are more significant for 6 and 8 m/s.

We will add text to explain that the simulation must be "spun-up" for enough time for the wake of the front turbine to propogate to the back turbine and that, for the 32 wind turbine case, the 2,000 seconds were necessary for this propogation to be achieved for the 6 and 8 m/s inflow cases.

Further clarification added: Line 249- "This was again done out of an abundance of caution---these lower wind speed cases did not quite fully develop after 1,000 seconds, so 2,000 seconds are discarded, and the remaining 1,000 seconds are examined."

16. In section 3.3, it is not clear what wind characteristics are besides an annual mean wind speed.

In the wind farm analysis the full wind rose is represented. There are therefore directional proportions, a Weibull distribution in each sector, and simulations performed across wind speed values from 4 to 24 m/s. A constant turbulence intensity of 5% and a power law shear coefficient of 0.2. We will consider whether any of the above information could be better highlighted in this section.

Section 3.3 already includes the following information in this regard: "The site is assumed to have the same Weibull wind speed distribution for each inflow direction, with k=2 and an annual mean wind speed of 10 m/s. Simulations were undertaken for a total of 72 inflow directions (5 degree direction bins). For each inflow direction, simulations were performed for hub-height ambient mean wind speeds from 6 m/s to 24 m/s, in 2 m/s increments." We did

however neglect to specify wind shear, and so have added "A power-law vertical wind shear profile was applied throughout, with shear exponent 0.2". Thank you for pointing out this missing detail.

17. In Fig. 2b, a matrix of 32 turbines is considered in a symmetric pattern. Because of a symmetric pattern, even with consideration of different flow angles, it seems 16 turbines are enough. Please explain more about using 32 wind turbines and justify this assumption.

The TotalControl wind farm is a standard benchmark that we did not invent. While there are certainly symmetries present, which might allow for computational cost savings in some contexts, we do not see how one could obtain the same results as in our reported analysis using only 16 wind turbines – especially given that we simulate about the full wind rose. Outside of possible computational savings, we do not see further benefit from seeking to reduce the number of turbines simulated.

For the outlined reasons, no changes have been made in response to this comment.

18. In Fig. 2b, there is no information about the arrangement and distances of the turbines. Also, out of curiosity, is there any reason to start turbine numbers from 0? It is suggested to add a table of turbine coordinates and naming conventions to make the layout traceable.

The utilised wind farm is a literature reference farm, the TotalControl standard wind farm, which has a standard definition (including arrangement, distances between all turbines etc) available via the reference provided in the text. We will check to ensure the standard nature of the utilised wind farm, and the link to its data and info, are prominent in the text of this section – along with key wind farm info such as turbine separation in x and y.

We have added an additional reference in the figure caption (Now Fig. 3b), but otherwise we feel these points are clear.

19. In line 215, equal weighing is used in rating life for turbulent wind model conditions. How do the authors justify such an assumption by considering variability in shaft speed due to the turbulence regime?

Combining across turbulent seeds is the second step in the process of combining rating lives. Step 1 was to turn each individual simulation into a single resultant rating life. It is in that step that rotational speed variations are accounted for, by converting each L10 life to units of time (accounting for rotational speed) and then combining. At the level of turbulent seeds, we are accounting for natural variability in the resultant rating life arising for any individual simulation at a given mean wind speed. It follows that we combine these resultant lives using equal weightings, since none occurs "more" than the others in this context.

For the reasons outlined here, no change to the manuscript was made in response to this comment.

20. According to Fig. 4, the life results for the bearings for both turbines are in a different order. How do the authors describe such a phenomenon? Is the assumption of the taking loads by the front bearing reasonable? It should be noted that the bearings have the same order of radial dynamic load rating, and when the life has a different order, it shows a different order of loads.

We assume you are referring to the relative difference in rating life between the two main bearings for the unwaked front turbine. This is driven by the fact that the rotor-side bearing

reacts axial and radial loads, and the generator-side bearing only reacts radial loads. As a result the upwind bearing sees considerably higher loads, and also loading with differing qualitative characteristics due to the design of the turbine aerodynamic thrust curve (which peaks near where rated power is first reached). Yes, axial load reaction by the front bearing is one of the possible configurations which has been applied in practice. Note, due to effects related to thermal shaft expansion, only one of the main bearings may be axially supporting. Concerning your final point, yes we agree, and as above this stems from the rotor-side bearing seeing both axial and radial loads, with the former being significant.

Clarification on this point has been added to the paper: Line 281-" Differences are also observable between wake impacts seen for the rotor-side versus generator-side main bearing, due to the former reacting axial and radial loads and the latter reacting radial loads only."

21. Interestingly, the trend of the basic rating life of the bearing in the rotor and generator side regarding the wind speed in front turbine is different. The life of the rotor side at 7.5 m/s is higher than 11 m/s. On the other hand, on the generator side, the bearing life at 7.5 m/s is almost half of the other wind speeds. It would be valuable if the authors discussed more about this happening.

This effect you describe appears (unless we're looking at the wrong part of the plot) to only manifest for large value of Back Turbine Offset. In such cases there is little or no wake impacting the downstream turbine, and we can see that in those cases the results tend towards those seen for the front (unwaked) turbine. Hence it seems this is again an observation of your observation in point 20, above. We will consider if any of these effects would benefit from further discussion in the manuscript.

As this behaviour is simply a convergence to unwaked bearing life results (show in the same plot) for large turbine offsets (greater than one rotor diameter), we didn't feel there was much to be added into the discussion on this matter.

22. The results in Fig. 4 show the difference between the life of the bearings in front and back turbines. It would be useful to present the average power in the same figure, as maybe less power is one of the reasons behind the shorter life of the bearings.

We will consider whether power results here might provide valuable additional context to this figure. It might help highlight that there is a trade-off here between increased power capture under partial waking (relative to a full wake) and main bearing design lives. We don't quite follow your second point, and can't see why reduced power might lead to a shorter bearing life? Irrespective of this, we'll consider these points and seek to enhance the discussion of these results in the manuscript.

We checked and didn't find that the addition of power in the plots adds anything to help the discussion and analysis here, so we have kept the figure as it originally appeared. As indicated above, we're not sure why less power would intrinsically link to a shorter bearing life.

23. In line 257, the standard grid spacing claim needs a reference.

We'll add a source for this, thanks for point out that there currently isn't one.

We have now added a source for this.

24. Please add the illustration of the wind rose used in Fig. 5a. It can be added in Fig.3.

This is a great suggestion! We'll certainly do this.

This has now been done (please see what is now Fig. 4).

25. In line 267, it is observed that the basic rating life always decreases in the bearing, while in a few specific conditions in the parametric study, the life increases. To have a fair claim, the condition of the turbine in spacing and wind conditions should be the same.

We feel this is a reasonable claim to make based on what we have shown in the paper, but perhaps we can improve on the wording a little… we'll revise this sentence to read: "Therefore, while it has been shown that some specific conditions can result in rating life increases (see Fig. 4), the more commonly observed life-reducing cases appear to dominate overall"

This has been done as described.

26. Typing error in line 37: Redundant Kenworthy et al.

Thanks, this has now been sorted.

wes-2025-63: Reductions in wind farm main bearing rating lives resulting from wake impingement

**Response to Reviewer 2**

We thank the reviewer for their comments, which we feel have helped improved the quality and clarity of this manuscript. Reviewer comments are listed below, followed by our responses in blue.

Introduction
• Lines 35-43: Two questions are stated – one related to validity of ISO-based main bearing rating life, and one related to what constitutes a realistic system model. I assume that this research is an attempt to answer the latter, but this is not very clearly stated. Am I correct? If not, the second question seems redundant. Please rephrase.

The current study is essentially seeking to (at least partially) address both questions, with the outcome shedding light on both whether wake effects are required for a sufficient system representation, and to what extent this might allow for ISO bearing life equations account for reported rates of field failures. We agree, however, that this could be better explained at this point of the manuscript, and so we will improve this discussion when undertaking revisions.

The sentence in question has been revised to read: Line 42-"The current work seeks to contribute towards addressing both of the above questions, by considering…"

• Line 37: Duplicate reference to Kenworthy et al.

Thanks, we'll sort that.

Sorted.

• Lines 44-46: Inconsistent use of "Sect." and "Section" – please check the guidelines of WES.

We'll confirm the correct style and update the manuscript accordingly.

Sorted.

Background
• The paper "Main bearing response in a waked 15-MW floating wind turbine in below-rated conditions" by Krathe et al looked at partial wake impingement effects on main bearing rating lives and should be referenced here.
https://link.springer.com/article/10.1007/s10010-025-00808-z

Thank you for flagging this paper to us. We agree it is a relevant reference to discuss here, and we'll add this in when revising the paper.

Proper consideration of this additional reference has now been added to the paper, please see Lines 116-124, 350 and 358 in the revised manuscript.

Section 2.1
• Line 59: Radial and axial bearing loads are referred to here but not defined until p. 7. Please check give a brief description of them here.

Will do.

On reflection, the material in Section 2.1 is bearing-application agnostic. Therefore, we feel it is not necessary to explicitly state the form of the axial and radial bearing load for a wind turbine main bearing at this stage in the paper.

Section 2.2
• Lines 109-110: The fatigue damage of the bearings depend highly on CD, and it is not useful to compare the damage of the upwind and downwind main bearing without commenting on the difference in CD.

Valid point, we'll add that context to this discussion.

This information has now been added: Line 110-" RCF life consumption based on ISO 281 was found to be fastest for the upwind bearing by 2 orders of magnitude, while the upwind bearing $C\_D$ value was only 23% greater than that of the downwind bearing, in their four-point drivetrain."

Section 2.3
• Line 131: The reference applied for the Dynamiks Python package looks strange. Please check that it is presented as intended.

Will do.

Fixed.

• Line 144: For someone not familiar with the model proposed by Hart, it is not trivial to understand what the elliptical and folding parameters describe. Please provide a brief explanation indicating what physical properties these parameters describe. In general, a more detailed description of this method would be useful to understand the results of this work.

Fair point, we'll expand on the description of this model and seek to provide a more complete and intuitive explanation.

An improved description has now been added: Line 154-" A parametric model for describing the distribution of wind direction at a site has recently been proposed (Hart, 2025). The parametric model utilises ellipses-of-unit-area to specify wind roses, with the probability associated with any direction-segment being equal to that segment's area. This generalized elliptical wind direction rose has three parameters: a prevailing wind direction, elliptical parameter and folding parameter. Restricting the ellipse to be of unit area results in a single parameter, a, which determines the shape of the baseline ellipse, from circular to increasingly elongated. The folding parameter then specifies a proportion (from 0 to 1) of ellipse probability mass to be ``folded'' across the minor-axis, thereby establishing a chosen level of bi- versus uni-directionality. Finally, the resulting wind rose model is rotated to obtain the specified prevailing wind direction."

Section 3.1
• Please provide more details related to the turbine. A table summarizing rated wind speed, hub-height, shaft tilt and rotor diameter would be useful. Is this a geared drivetrain?

Thank you for this suggestion. We will include a new table with relevant turbine info for the 10MW DTU turbine. Yes, it is a medium speed geared drivetrain (we'll highlight that also).

On reflection we felt that a further additional table would start to make the paper cluttered. These value shave therefore been added in the relevant paragraph when the 10 MW turbine is introduced. Please see Line 183 onwards.

• I assume that the wind farm is landbased (not offshore), but this is not stated anywhere. Please clarify.

The TotalControl reference wind farm is offshore, hence we did not simulate any terrain effects and used a 5% TI, we'll add a clarification to the paper.

Clarification added: Line 242- "Since the site is conceived of as being offshore, there are no modelled terrain effects."

• What is the rationale behind the choice of 5 % turbulence intensity? Turbulence intensity will significantly influence the wake recovery, which could alter the conclusions of this work. For a landbased turbine, 5 % is quite low compared to values recommended in the standards. It is important to discuss the validity of this assumption. The paper seeks to explain premature failure in main bearings, mainly reported for landbased turbines. If turbulence intensity is generally higher than what applied in this work, so that the wake recovers more quickly, it might not be valid to conclude that farm effects contribute that much to reduced main bearing lives.

We used 5% TI as a representative value for some offshore sites. For example, the following figure shows TI values from a mesoscale simulation of the North sea (based on data from https://orbit.dtu.dk/en/publications/environmental-mapping-and-screening-of-the-offshore-wind-potential):

[Figure]

Similarly, the following analysis of offshore sites shows that 5% TI is around the mode of TI distributions for various offshore sites, and again a reasonable representative value. https://windeurope.org/summit2016/conference/allposters/PO293.pdf

It is a valid point, however, that for sites with higher TI the wakes may recover faster, possibly lessening the impacts we report here. We'll therefore highlight this when revising the manuscript.

The Wind Europe reference has been added to motivate the chosen level of turbulence (5%) as a low-to-moderate offshore TI value for the study. We now also point out that faster wake recovery at higher TI sites might reduce wake impacts on the MB for those sites: Line 368-" Since all simulations were undertaken for a low-to-moderate level of turbulence (5% TI), it is

also possible that the magnitude of observed wake impacts would be reduced in higher turbulence conditions where wake recovery is faster"

• How is shear modeled in this work? If the power-law is applied, what shear exponent is used? Wind shear is highly important for main bearing rating lives. Combined with the wake deficit, the shear profile will determine what the "final" shear that the downstream turbine experiences. I.e. low shear could result in a "reversed" shear profile in which the wake velocity deficit (which is typically deflected vertically due to shaft tilt) leads to reduced mean wind velocity with height. Please clarify and discuss.

We used a power law shear profile using a shear coefficient of 0.2. We'll make sure this is clearly indicated in the paper, and will add some discussion about the impacts of this assumption and suggest possible future work looking into these interactions in detail.

We have now clarified that: Line 247-"A power-law vertical wind shear profile was applied throughout, with shear exponent 0.2". Concerning the more complex topic of evolving shear profiles as inflow passes through one turbine and impinges another, we have decided this falls outside of the immediate scope of the current paper. This is because a detailed consideration of these effects would require a more sophisticated model for vertical shear, and likely a direct analysis using large eddy simulation. Furthermore, without having access to those models and results we'd largely be relying on conjecture to say much about this topic. We agree this is an important consideration and will seek to build it into our future work, but we feel that a detailed discussion in the current paper would risk straying from our main focus – an analysis of main bearing rating lives using the developed tool chain.

• It is common in industry to use a generator-side locating (carrying axial loads) bearing. To be relevant for industry, I would recommend reversing the setup (I assume this does not require running Dynamiks simulations over again but is related to post-processing).

We cannot make this change without needing to re-run all simulations, as a result of the manner in which results were obtained. In addition, there are a variety of drivetrain setups which exist that include both configurations. Either way, we feel that our analysis is valuable and instructive to both cases. The setup we utilised here was based on a benchmarking drivetrain designed by ORE Catapult in a commercial project.

For the above reasons we have not made any changes to the manuscript in response to this comment.

• Please state what X and Y (load factors) are applied for each bearing. This is useful to understand the importance of thrust versus radial loads in the fatigue calculations.

This can change based on the ratio of axial radial bearing load, we'll add some more info to clarify what the values and change point are.

This information has now been added, please see Line 211 onwards.

• What is the rationale behind the choices of CD? Are these values representative of 10 MW turbines? The authors later (Section 4) comment on the high rating lives, but these results highly depend on CD.

Bearing data for large wind turbines are not easily obtained, and aren't available in the literature. For instance in https://doi.org/10.1002/we.2476 main bearings are specified which

don't meet the design life for that turbine, and apart from this there are little to no alternatives. The main bearing specs used in this study were obtained via ORE Catapult, from a drivetrain they designed for a 10MW wind turbine within a commercial benchmarking project. We'll add some clarification of this to the manuscript. It is also possible to show that results scale directly with CD (due to linear properties of the resultant rating life equations), which generalises the results beyond any single bearing design. We'll include that discussion when updating the manuscript.

We have included some additional information on the bearing design specifications, as well as some context concerning where the bearing design came from. Most importantly, we have also shown why, for the relative results which are out main focus, results are not impacted by changes in CD. Please see Lines 213-217 in the revised manuscript.

• The authors investigate a 10 MW turbine, while main bearing failure reports mainly exist for smaller turbines. Could the authors comment on whether wake effects can be generalized across turbine sizes? Could wake effects be less important for smaller turbines, and therefore not have result in the same reduction in main bearing rating lives?

We'll happily include some discussion of these questions. Generally speaking we'd expect wake effects in the context of main bearings to be fairly general across turbine scales, given that turbine spacing tends to scale with the size of the turbines. While wake effects could conceivably be less important for smaller turbines, there remains a gap between predicted and observed main bearing lives across all scales for which, from the current work, wake impacts appear to be at least a credible candidate contributing cause.

The following has been added to the Discussion in order to highlight turbine scale as a factor here: Line 370-"The present study was also undertaken using 10 MW wind turbine models; in future work it would therefore be beneficial to quantify how main bearing wake impacts manifest at different scales". While we have some thoughts about how wake impacts might manifest across scales, we felt these were not concrete enough to state in the paper. Therefore, we have simply ensured to highlight a future need to understand scaling effects on these results.

Section 3.2
• Why is 5D applied in the two-turbine parametric analysis?

We undertook the same analysis at 3D and 4D and all results were qualitatively the same, hence we avoided overburdening the reader with lots of matching results. We'll clarify this in the paper and include a reference for why 5D is a sensible separation to choose as the nominal separation distance.

We have added the explanatory text: Line 225-"We chose 5 rotor diameters as a quantity that is often examined (e.g., in Simley et al. (2020a)) and similar geometries have been used when validating novel control strategies (e.g. Simley et al. (2020b)).

Section 3.3
• It could be useful to put the parameters presented here (e.g. k, annual mean wind speed, mean wind speeds, inflow directions etc.) into a table for better overview.

Good idea, we'll do that!

On reflection, all of these details are provided together at the start of Section 3.3. We therefore don't feel that much will be added by also summarising in a table. Happy to reconsider if you feel strongly about this, but we didn't want to make the paper longer unnecessarily.

• What is the spatial grid resolution in the wind farm simulations and turbulent wind fields?

We will clarify in the paper that our turbulent wind field has a resolution of 3 meters in x, y, and z dimensions. Additionally, each turbine wake has a turbulence box with (3.8, 1.5, 1.5) m resolution in x, y, and z, which is the default setup in Dynamiks.

These details are now given in Table 2 of the revised manuscript.

• Line 214: Suggest rephrasing to: "For each main bearing and each direction ":

Agreed.

Changed to: " For each main bearing within the wind farm and each wind direction"

• The distances between turbines along x and y should be stated more clearly

We'll clarify that the TotalControl wind farm has east-west spacing of 10 D and north-south spacing of 5 D.

This has been added on line 242.

• Figure 2a: Axes missing.

This subfigure is indicating the setup for the parametric analysis, rather than seeking to provide full positional data, and all pertinent info is stated in the caption and/or manuscript in non-dimensional nD form.

For the reasons outlined we've not made any changes in response to this comment.

• Line 221: The reference to Hart should not be in parenthesis.

Good spot!

Sorted.

• Line 221: "Based on model fitting to data" – what kind of data? For what location are these ranges of wind roses realistic? What are the criteria for realistic? Is this data site-specific?

Will add some additional context here, thanks for highlighting this.

Clarification has been added that data fitting occurred for data from offshore wind farms. Note, all of the data used is available via the linked wind rose modelling article.

• Figure 3: Do all the wind roses evaluated have the prevailing direction of 180 degrees?

The simulated TotalControl wind farm is designed for a site with prevailing wind direction along the East-West direction (see Figure 2b), and we therefore maintained that prevailing wind

direction throughout our analysis. More generally, the modelled wind roses can be readily rotated in order to shift to a different prevailing wind direction, but this didn't seem appropriate for the current analysis.

This has been clarified in the manuscript: ?-" The prevailing (highest probability) wind direction corresponded throughout to the left-to-right flow direction apparent in Fig. 2b." Additionally, we realised that the axes used for wind rose illustration in Fig. 3 didn't conform to the direction frame used in the TotalControl wind farm. We have now corrected the polar plots to have the correct zero point and positive direction.

Section 4
• It could be useful to split this section into subsections to have a better overview of the different results.

We'll gladly consider whether this approach might improve readability.

We have now split Results and discussion into two separate sections for Results and then Discussion. Results is further split into two subsection. We agree this helps readability, thank you for the suggestion.

• Line 228: The authors assume that the locating main bearing fails most commonly. What is this assumption based on? Why not present results of the rear bearing too (e.g. in the appendix)?

The locating bearing failing most commonly is established based on failure data, and discussions with industry. We'll add a citation. Given the weak link is the locating bearing, it seemed to be distracting to overly focus on the rear bearing results as well. We'll revisit this decision when revising the manuscript to see if there could be merit in including these additional results.

We still feel strongly that the focus should be on the bearing which fails most commonly (the locating main bearing). We've added a reference to support that being the case.

• Line 230 and 240: "...bearing rating lives can be seen to far exceed the minimum design life..." Again, the rating lives are dependent on the value of CD. A more detailed description of the choice of CD should be given if these findings should be considered important.

We'll include that improved context here, but it's also worth highlighting that these findings are consistent with our previous paper (https://doi.org/10.1002/we.2883) in which these same findings held for bearings where we had full bearing data for commercially available main bearings. Certainly we can reference this earlier work to shore-up this discussion.

The key point here is that main bearing can be readily designed with a sufficiently high CD value, according to ISO 281 or 16281, but they still experience large numbers of premature failures. Therefore, the key point here is more nuanced than simply considering CD in isolation. We have improved the discussion at this point on the apper by including the following:Line 270-" Considering the (unwaked) front turbine results first, rotor- and generator-side bearing rating lives can be seen to far exceed the minimum design life of 20 years, as would be expected based on previous work (Kenworthy et al., 2024). This again highlights the key domain-challenge presented by main bearings which have a sufficiently large dynamic capacity, CD, according to rating life formulations (Kenworthy et al., 2024), but which still failure prematurely in large numbers (Hart et al., 2023; EPRI, 2024)."

• Figure 4: Asymmetries are more pronounced for higher wind speeds. Could the

authors comment on the differences in results between wind speeds?

This is a good point, and could be resulting from a variety of interacting effects. Perhaps the most obvious candidate would be that aerodynamic loads are more sensitive to wind perturbation at higher wind speeds (due to the v^2 term in lift and drag loads). Similar changes in local wind speeds due to wake presence could therefore be expected to elicit a greater magnitude of response in higher wind-speed cases.

This observation is now included as follows: Line 279-" The observable asymmetry also becomes more pronounced at higher wind speeds, possibly as a result of increased aerodynamic sensitivities stemming from wind-speed-squared lift and drag terms"

• Lines 245-255: I think this explanation of the asymmetry is a bit too simple. Gravity mainly acts in the in-plane-bending moment in the blades and less so in the out-of-plane bending moment, depending on the shaft tilt and curvature of the blades. Out-of-plane blade root bending moments are predominantly important for main bearing loads (relative to in-plane BM). Is gravity in the blades driving hub pitch and yaw moments? When removing gravity, as presented in Fig. A1, the shaft moment due to rotor weight vanishes, and the radial loads are significantly reduced. With regards to the locating main bearing, bearing rating lives are now likely governed by axial loads, so that any asymmetry trend would disappear among the axial loads. It would be interesting to see a closer investigation of this effect.

We respectfully disagree on this point. The fact that turning gravity off removes this asymmetry in results provides strong evidence for the position that the effect relates to rotor in-plane forcing effects, which can interact with the gravity force vector. Importantly, all in-plane moments along the blade are generated by in-plane forces acting at a distance from the centre of rotation. It is those forces which can then act to perturb the mean vertical force being applied at the hub centre. Even if these changes are relatively small at the force level, recall that bearing rating life scales as 1/(applied load)^10/3, hence any change in applied vertical forcing will be magnified by this effect.

Note, we agree with your argument that the loading situation is very much changed once gravity is turned off, but gravity (as you point out) doesn't impact out-of-plane rotor forces and moments very much at all. If the asymmetry in results was principally driven by changes in out-of-plane moments, then we'd expect that asymmetry to persist in the absence of gravity. As that's not what we see, we conclude that it must be an effect of the type outlined above driving these interactions.

For the above reasons we have not made changes in the paper in response to this comment.

• Line 257-258: "Within a wind farm, the standard grid spacing between turbines will commonly be on the order of 3D-5D". Is this referring to spacing in the predominant cross-wind direction? I believe that larger distances are seen in the predominant wind direction. A reference would be useful.

You are of course correct on this point, and this is also the case for the wind farm we simulate in our study. We will correct this and provide a relevant reference as you suggest.

We have corrected this to read: Line 304-" Within a wind farm, the standard grid spacing between turbines will commonly be on the order of 6D-10D in the prevailing wind direction, and 3D-5D in the cross-wind direction (Manwell et al. 2010). As the wind direction changes the cross-flow offset between turbines (relative to the inflow direction) will vary continuously across the full range analysed in this parametric analysis and beyond."

• Lines 275-285: Again, it would be useful to explain the physical meaning of a and f before discussing their impact on main bearing rating lives.

Fair point, will do!

A proper description of these parameters and their impact on the wind rose was included in response to an earlier comment. Please see Line 154 onwards of the revised manuscript.

Conclusion
• The impact of turbulence intensity and shear on the results should be discussed.

We will add in discussion of these important points, and highlight that more detailed investigations into their impacts in this context will be important future work.

The conclusion section highlights the need for all results to be carefully interpreted in light of the modelling simplifications which are present. This of course includes both turbulence intensity and shear characterisations, and both of those items have been better treated in the manuscript itself (in response to above comments). However, the conclusion section doesn't aim to unpack all of the various limitations again, but rather highlights that they are present. As such we don't feel it is appropriate to include explicit mention of TI and shear as particular limitations in the conclusions, as this would indicate they are more important than others which are not mentioned. Therefore, while we have gladly implemented revisions based on earlier comments regarding both TI and shear, we have decided not to give them further mention in the Conclusion section.

---

## Referee Report (RR1)

**Minor Comments**

I would like to thank the authors for their thorough responses and the revisions made to the manuscript. The quality and clarity of the paper have clearly improved. However, a few minor issues remain that should be addressed prior to final acceptance:

**Section 2.1**

While the authors correctly noted that the bearing life calculation ($L_{10}$) is based on revolutions, this is still not explicitly stated in the text. To ensure clarity, it is recommended to revise the sentence *"is the proportion of time spent in the ith set of conditions"* to *"is the proportion of the total operation that occurred under operating condition i."*

**Section 3.1**
**Main bearing load estimation**

It remains unclear whether the loads are applied at the *center of the hub* or at the *interface between the hub and the main shaft*. If the loads are indeed applied at the hub center (as suggested by Figure 2), the schematic of the drivetrain should be updated to include the relevant dimensions of the main shaft, as well as the distance between the front bearing and the hub–shaft intersection.

**Main bearing rating life assessment**

In Table 1, the *pitch diameter* of the bearing is reported, but this parameter is not introduced or defined in the manuscript. Please include a brief explanation or definition where it first appears.

**Section 3.2**

In their response, the authors indicated that the *wind characteristics* would be summarized in a table. This information should be included in the revised manuscript to improve completeness and transparency.

---

## Author Response (AR2)

The authors thank the reviewer for their additional comments. These final issues have now all been addressed as follows:

Section 2.1 While the authors correctly noted that the bearing life calculation ($L10$) is based on revolutions, this is still not explicitly stated in the text. To ensure clarity, it is recommended to revise the sentence "is the proportion of time spent in the ith set of conditions" to "is the proportion of the total operation that occurred under operating condition i."

This change has been made as suggested, thanks!

Section 3.1 Main bearing load estimation It remains unclear whether the loads are applied at the center of the hub or at the interface between the hub and the main shaft. If the loads are indeed applied at the hub center (as suggested by Figure 2), the schematic of the drivetrain should be updated to include the relevant dimensions of the main shaft, as well as the distance between the front bearing and the hub–shaft intersection

The loads are applied at the center of the hub. Figure 2 already includes all relevant dimensions (Lh and Lb) for calculating the force response at each main bearing (see Eqns 4-9). Other main shaft dimension do not impact this load balance and so are not required to ensure readers are able to recreate the analysis.

Main bearing rating life assessment In Table 1, the pitch diameter of the bearing is reported, but this parameter is not introduced or defined in the manuscript. Please include a brief explanation or definition where it first appears.

Good point, a description of what the pitch diameter is has been added as requested.

Section 3.2 In their response, the authors indicated that the wind characteristics would be summarized in a table. This information should be included in the revised manuscript to improve completeness and transparency

Table 2 has been expanded to include the relevant wind characteristic information for the study. Thanks!